# All-plasmonic sub-terahertz wireless communication link

Tobias Blatter [1,3] ✉, Stefan M. Koepfli [1,3] ✉, Amane Zuerrer[1], Samuel Hess[1], Yannik Horst[1], Marcel Destraz[2], Daniel Rieben[1], Michael Baumann[1], Laurenz Kulmer[1], Jasmin Smajic[1], Yuriy Fedoryshyn[1] & Juerg Leuthold[1] ✉

A promising approach to increase wireless capacity is the transition to sub-Terahertz carrier frequencies (0.1–0.3 THz). While traditional high-frequency approaches employ III-V semiconductor technologies, plasmonics is emerging as a potential solution for highest-speed components. In this paper, we introduce an all-plasmonic sub-THz wireless link, utilizing compact (<50 μm²) plasmonic components that exhibit a flat frequency response up to 300 GHz while providing full flexibility in carrier frequency selection. The plasmonic approach offers unprecedented integration potential, compatibility with diverse platforms, and scalable, cost-effective fabrication. To demonstrate its capabilities, we conduct a lab experiment transmitting 120 Gbit/s on a 285 GHz carrier across a 5 m free-space link, validating the system's linear performance and large power dynamic range. While this first demonstration is constrained in transmission distance, it showcases the transformative potential of plasmonic technology in closing the wireless-optical data-rate bottleneck: The proposed plasmonic converters could provide the capacity expansion needed for future 5G, 6G, and beyond wireless networks, paving the way for high-speed, cost-effective, and scalable sub-THz communications.

Optical and wireless communications networks are at the core of our society and will further shape our future[1,2]. Fiber-based optical transmission is the backbone of communication, already achieving capacities in the hundreds of terabits per second[3]. Wireless radio frequency (RF) links provide essential point-to-point and point-to-multipoint connectivity, particularly for bridging last-mile gaps, extending networks to underserved regions, and enabling mobile communications. Yet, the capacity of wireless links lags behind. They are unable to meet the projected demands of next-generation communication standards such as 6G and beyond[4,5]. The capacity discrepancy between fiber and wireless networks also poses a problem in view of a convergence of the two networks. In many situations, high costs, logistical challenges, and impracticality of deploying fiber in remote or mobile environments necessitate a fiber-wireless-fiber solution. The ideal wireless channel should offer capacities in excess of 100 Gbit/s – today's single channel capacity to ensure a seamless convergence with fiber. Unlike free-space optical solutions, next-generation wireless links should be resilient to adverse weather conditions such as fog, haze, and atmospheric turbulence[6–11]. They should be based on an affordable technology and offer a compact footprint in order to ease a widespread deployment[12].

As a solution to increase the wireless capacity, RF carrier frequencies in the sub-Terahertz (sub-THz) regime (0.1–0.3 THz) have been suggested. At higher frequencies, significantly larger bandwidths are available[4,13]. Two main transparency windows are particularly relevant: the 100–170 GHz and 200–330 GHz bands[14]. The first window is more accessible using electronic solutions and has already enabled successful high-data-rate and long-distance transmission links[15–18]. The second transparency window offers higher bandwidth, and transmissions in excess of 100 Gbit/s per polarization and per channel have been realized[19–25]. Even longer distances of 4.6 km at 24 Gbit/s[17] and 1.4 km at 184 Gbit/s[26] have already been shown.

[1]ETH Zurich, Institute of Electromagnetic Fields (IEF), Zurich, Switzerland. [2]Polariton Technologies AG, Adliswil, Switzerland. [3]These authors contributed equally: Tobias Blatter, Stefan M. Koepfli. ✉e-mail: tobias.blatter@ief.ee.ethz.ch; stefan.koepfli@ief.ee.ethz.ch; juerg.leuthold@ief.ee.ethz.ch

Towards the realization of sub-THz implementations, several competing approaches exist to generate and detect such high carrier frequencies[4,27–29].

Traditional all-electronic approaches rely on frequency multiplier chains[30] or resonant tunneling diodes[31] to generate sub-THz carriers, while high-electron-mobility transistors[32,33] and Schottky barrier diodes[34] serve as receivers. However, these systems face limitations such as narrow intermediate frequency bandwidths (in the order of tens of GHz), non-linearities, and non-flat frequency responses, which degrade signal quality and limit achievable data rates[35,36]. In addition, electronic approaches require an additional device for optoelectronic (OE) and electrooptic (EO) conversion when used to bridge optical receivers.

Using potent photonic OE and EO devices directly presents a more natural approach, carrying the potential to overcome electronic bottlenecks[19]. On the transmitter side, sub-THz signals can be photomixed by beating an optical carrier with an offset laser frequency. Uni-traveling carrier photodiodes (UTC-PDs) are commonly used to convert these optical signals into high-frequency electrical carriers, offering high bandwidth and relatively good output power[37]. At the receiver, electrooptic (EO) modulators can upconvert sub-THz signals back into the optical domain and send them to an optical receiver. However, traditional photonic modulators are bandwidth-limited (typically <100 GHz), restricting their ability to handle sub-THz carriers[38]. At the transmitter, UTC-PDs offer sufficient bandwidth. However, they rely on III-V semiconductor layers, which are expensive, difficult to integrate with silicon-based platforms, and challenging for large-scale production[39]. Although some high-bandwidth Ge-based and graphene-based devices have been demonstrated, UTC PDs are still the main solution, as they either provide small bandwidths or small carrier frequencies[40,41].

In the last few years, plasmonics has increasingly been emerging as a solution to highest speed. On the EO side, advancements in plasmonic modulators have shown to be able to operate within the sub-THz with bandwidths beyond 500 GHz[42], and recently, operation up to 1 THz has also been verified[43]. They're energy efficient and are compact in footprint[44,45]. Recently, plasmonic photodetectors (PDs) based on graphene have been shown to reach bandwidths beyond 400 GHz[46] and even 500 GHz[47]. With this, the OE side is matching the speed of the EO side and could enable sub-THz wireless communication. However, plasmonic graphene PDs have not been explored or verified for sub-THz communication due to a lack of simultaneously achieving sufficient output powers and operation in the 100 s of GHz.

In this paper, we demonstrate an all-plasmonic sub-THz wireless link, utilizing plasmonic photodetectors and modulators to enable high data rate, high-frequency communication. This plasmonic architecture offers flexible carrier frequency selection, chip-scale integration on silicon, and a broad linear power dynamic range, addressing the scalability and cost challenges of conventional III-V and electronic technologies. To validate this approach, we implemented a 285 GHz wireless link, achieving 120 Gbit/s over a 5-m transmission distance. Our results highlight the potential of all-plasmonic technology for future high-speed, low-power, and cost-effective wireless-optical-wireless networks, paving the way for next-generation 5G, 6G and beyond.

## Results
### Vision of all-plasmonic sub-THz link
The vision of an all-plasmonic fiber-wireless-fiber link is illustrated in Fig. 1. A remote data center sends data via an established optical fiber communication channel. The fiber network is locally interrupted, e.g., as represented by the river in the figure. Here, the optical signal can be converted into the sub-THz frequency regime where sufficient capacity is available and transmission in free space is possible. After bridging the obstacle, conversion back to the optical fiber network would

create a seamless transition between the drastically different carrier frequencies.

To convert the signal, we envision the use of plasmonic optoelectronic (OE) and electro-optic (EO) components, as represented by the blow-out representations in Fig. 1. A plasmonic photodetector is used as sub-THz source, whereas a plasmonic modulator is operated as sub-THz detector. The devices thereby form two remote antenna units (RAU) that connect the optical coherent transmitter (Tx) to a corresponding receiver (Rx). To verify that this vision is feasible, we build such a link in a laboratory environment. First, we verify the individual device performances. Afterward, we include the devices in the data transmission setup.

### Opto-electric conversion with plasmonic photodetectors
First, we discuss the plasmonic photodetector (PD) for optoelectronic (OE) conversion from the telecom wavelength to the RF domain. The plasmonic OE converter is based on a graphene-metamaterial architecture, as visualized in Fig. 2a. The architecture is similar to the one introduced in ref. 47. A metamaterial perfect absorber[48] layer stack in the form of a horizontal metal-insulator-metal is used. The top metal layer consists of dipole resonators that are interconnected with lines. The graphene bilayer is positioned directly underneath the resonators. The full structure is encapsulated with an aluminum oxide layer to enable stable operation in air. Detailed dimensions and details on the fabrication are provided in the Methods section.

We explain the physical operation principle that results from the above-described architecture. The plasmonic graphene PD operates in a photovoltaic (PV) mode, which is enabled by the metamaterial perfect absorber structure. Light impinging on the metamaterial structure leads to a plasmonic resonance at the resonator. The resonator is inductively coupled to the metallic backplane, which leads to a trapping of the light within the layer stack[49,50]. A fraction of the light is absorbed within the graphene layer close to the resonators. The absorbed photons generate free charge carriers. Due to contact doping at the resonators[51], a built-in field is induced, which drives the charge carriers to the collection electrodes and generates the PV photocurrent.

To understand and capture the PD's performance, we are performing DC electrical and low- and high-frequency electrooptic characterization. First, Fig. 2b shows the side view of the device under contact in our measurement setup. The device is operated as a graphene field-effect transistor. The gate is contacted with a DC needle probe. The drain-source channel is contacted by the RF probe. When sweeping the bias voltage $V_B$, the device shows a clear ohmic response, as plotted in Fig. 2c. When applying a gate voltage $V_G$ with the gate needle, we can modify the doping within the graphene channel and thereby its resistance and the shape of the built-in field. Figure 2d shows the gate sweep. The Dirac point is situated at −13.25 V. For further characterization, we will operate the device at zero bias voltage $V_B = 0$ V and at a gate voltage $V_G = -6.75$ V, where we found an optimal response. The gate dependence on the photoresponse is shown in the Supplementary Fig. 1.

To test the bandwidth of the device, we feed an optical signal to the device through a standard single-mode fiber (see also Fig. 2b). The optical signal consists of two laser signals with equal power and the corresponding RF frequency offset to generate the RF beating note. The setup for the bandwidth characterization is schematically illustrated in Fig. 2e. The electrical ground-signal-ground pads are probed with high-frequency probes and fed to an electrical spectrum analyzer (ESA). To access the frequencies above the measurement range of the ESA (i.e., 110 GHz), we employ sub-harmonic mixers in the 110–170 GHz and 220–330 GHz ranges. Figure 2f shows the recorded normalized RF power. The measurement shows no roll-off behavior in the measurement range, verifying that the PDs can operate across a broad range and are able to generate the targeted sub-THz carriers. The here

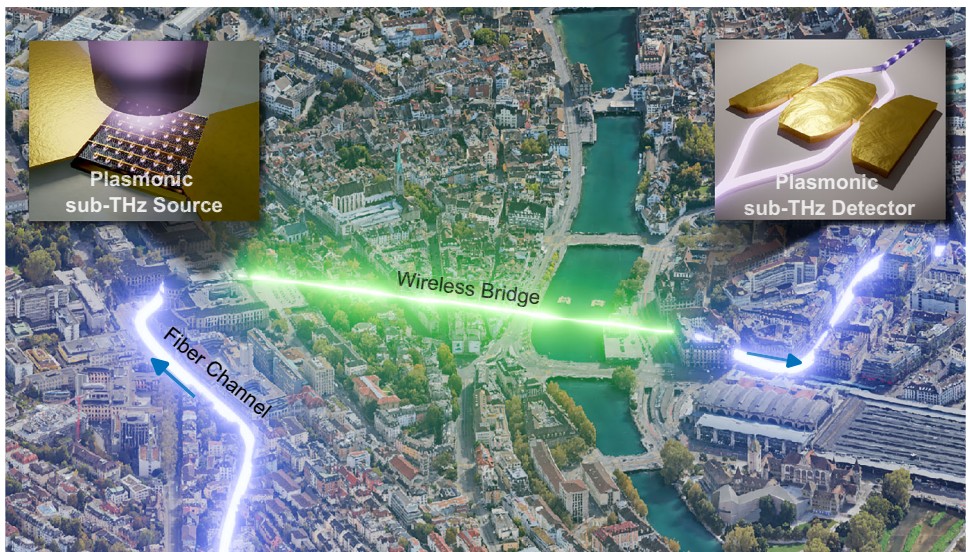

**Fig. 1 | Vision of the plasmonic-to-plasmonic wireless sub-THz bridge.** Data from a remote data center is sent via a fiber channel. To cross a gap in the fiber channel, the signal is converted to the sub-THz regime and back to the optical regime via the plasmonic converters.

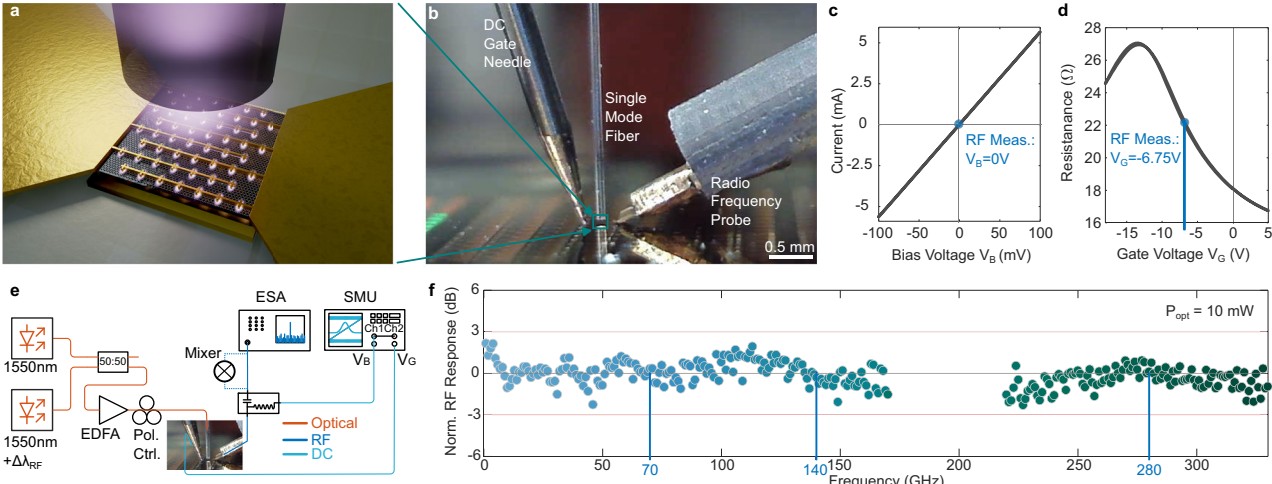

**Fig. 2 | High-speed OE converter: plasmonic photodetector characteristics.**
**a** Artistic visualization of the graphene plasmonic photodetector. The architecture consists of a gold back mirror and gate, an aluminum oxide spacer layer, a chemical vapor deposition grown bi-layer graphene sheet and a top layer consisting of interconnected plasmonic resonators. **b** Side view of the plasmonic photodetector in the measurement setup. The device is illuminated by a single-mode fiber from the top. A direct current (DC) gate needle is used to electrostatically dope the graphene. The radio frequency (RF) probe is used to read out the electrical signals generated by the photodetector. **c** I-V characteristic of the device. It shows a clear ohmic behavior. **d** Gate voltage sweep of the device resistance. The Dirac point is

situated at 13.25 V. The blue points in (**c** and **d**) mark the operation points for the following radio frequency measurements. **e** Schematic of the used laser beating setup for the photodetector characterization with two laser being combined in the optical path (orange), fed to an erbium-doped fiber amplifier (EDFA) which is then used to illuminate the photodetector. The RF signal (blue) is read out with an electrical spectrum analyzer (ESA) and the DC part (light blue) is read out with a source-measure unit (SMU). **f** RF response as a function of frequency. The photodetector shows a flat frequency response in the measurement range from 1 GHz to 330 GHz.

demonstrated >330 GHz bandwidth is limited by the characterization setup and >500 GHz could be possible as demonstrated in ref. 47, which is based on a similar architecture.

With the flat frequency response across a broad range, it is possible to generate on-demand carriers at frequencies within this range. To verify this property, we choose three frequencies: 70 GHz, 140 GHz and 280 GHz. First, we characterize the responsivity of the device across an optical input power range from 350 nW to 140 mW. The results are shown in Fig. 3a. For the broad power sweep, we use a lock-in amplifier and an on-off modulated optical signal at 320 kHz. The device shows a linear response (gray, circular data points) with optical power, resulting in a 6 mA/W responsivity. The photodetector is again

operated with a gate voltage of −6.75 V and without any DC source-drain bias.

Next, we again connect the plasmonic photodetector to the high-frequency measurement equipment with the laser beating illumination and perform additional power sweeps. The resulting RF power as a function of DC photocurrent is shown in Fig. 3b. The data points follow the expected 20 dB/dec characteristic and do not show any compression or saturation. We achieve an RF power output of around −37 dBm or 0.2 μW at 280 GHz. Furthermore, the output RF power remains the same across the three tested frequencies (70, 140 and 280 GHz), showcasing that the flat frequency response remains even when operating the device at higher optical input powers. This means

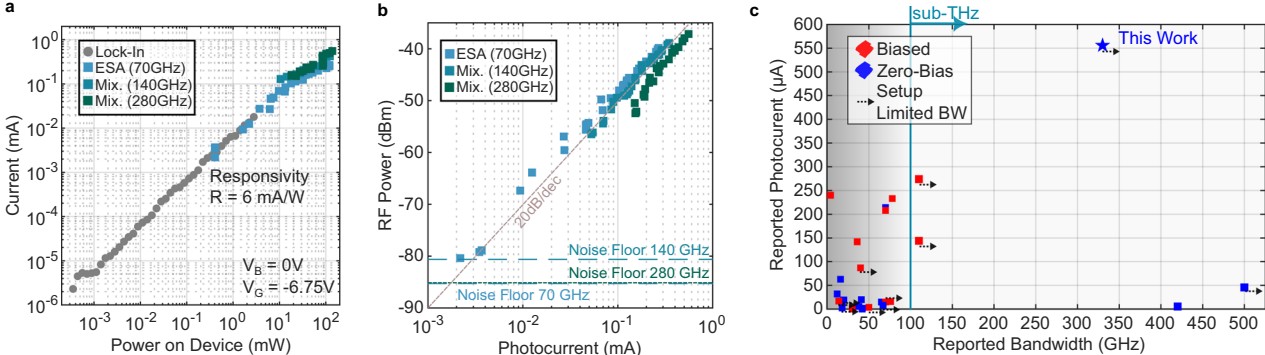

**Fig. 3 | DC photocurrent and RF power generation of plasmonic graphene photodetectors. a** Generated direct current (DC) photocurrent as function of optical power on the device. The slope leads to a responsivity of 6 mA/W. Gray dots correspond to measurements performed with a lock-in amplifier. Blue to green squares correspond to measurements where the PD was connected to the electrical spectrum analyzer (ESA) and sub-harmonic mixers. **b** Generated radio frequency (RF) power as function of DC photocurrent for three RF frequencies. 70 GHz: blue, 140 GHz blue-green, 280 GHz green. **c** Comparison of reported high-speed graphene photodetectors as function of achieved bandwidth and maximum output photocurrent. The red color corresponds to biased devices, whereas blue colors corresponds to zero-bias. The arrow indicated setup limited bandwidth (BW) measurements. Detailed parameters for each data point can be found in Supplementary Table 1.

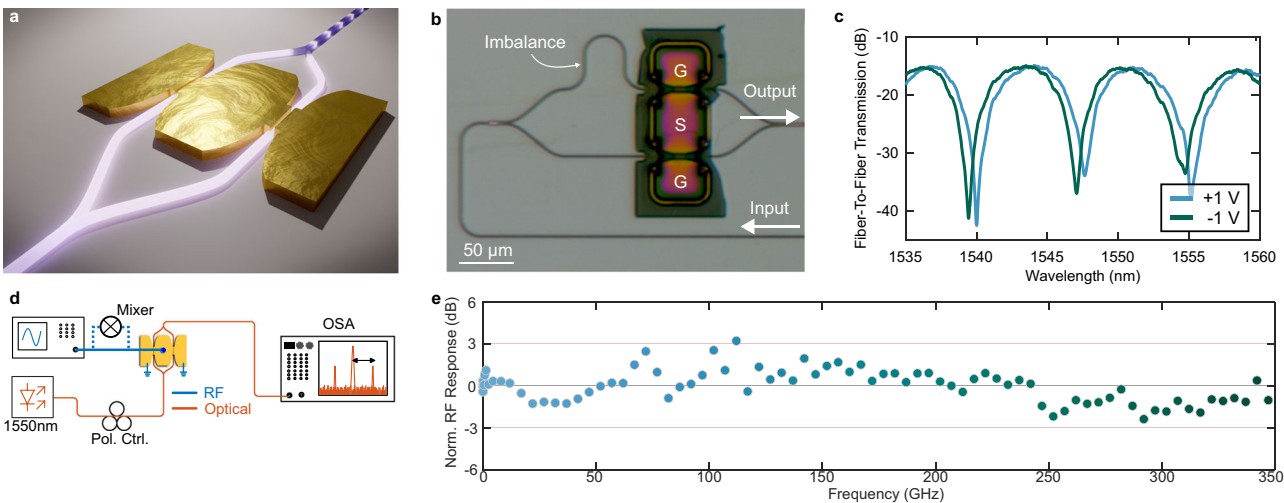

**Fig. 4 | Plasmonic modulator characteristics. a** Artistic representation of the plasmonic Mach-Zehnder modulator consisting of two plasmonic phase shifters. **b** Microscope image of the plasmonic modulator. **c** Measured fiber-to-fiber transmission when ±1 V is applied. **d** Schematic representation of the setup to characterize the electro-optic bandwidth of the plasmonic modulator. A radio frequency (RF) signal (blue) is used to drive the modulator. The generated sidebands on the optical carrier (orange) are detected using an optical spectrum analyzer (OSA). **e** Measured bandwidth of the plasmonic modulator in the frequency range up to 350 GHz.

the PD can flexibly generate various carrier frequencies on demand, enabling dynamic adaptation to system and channel conditions. For instance, under fog or snowfall, lower carrier frequencies benefit from reduced atmospheric absorption[52].

From the measurements shown in Fig. 3b, we also represent the points in Fig. 3a in the optical vs. photocurrent plot. We note that the range here is limited to the high optical powers, as the ESA and the sub-harmonic mixers introduce a varying leakage DC current onto the measurement channel, introducing uncertainties for the lower current regime. The points measured across the different measurement setups align, and no clear saturation is visible. The maximum extracted photocurrent is 0.556 mA, which, to the best of our knowledge, is the highest photocurrent reported for any high-speed zero-bias graphene photodetector as summarized in Fig. 3c. The changes that lead to the improvements in responsivity and maximum photocurrent compared to the prior work are further discussed in the Method section. With the here reported values, the device can be used in any scenario requiring high-speed infrared photodetectors, for example data

communication. Next, we will verify the usability of the photodetector as a sub-THz source in a wireless data transmission scenario with a carrier frequency of 285 GHz in the following sections after discussing the sub-THz receiver in the form of a plasmonic modulator.

## Electrooptic conversion with plasmonic modulators

In this section, we discuss the plasmonic modulator that is targeted to be used as a flat-response (sub-) THz detector. It consists of two plasmonic phase modulators forming a Mach-Zehnder modulator (MZM), as illustrated in Fig. 4a. The RF signal (S) is applied to the central electrode, while the two outer electrodes are grounded (G). Through the Pockels effect and the MZM configuration, the RF signal is directly mapped back to the optical amplitude. Concretely, the slot between the metal electrodes is filled with an organic Pockels material, creating two active regions where the optical and RF fields strongly interact[52,53]. In one active region, the light is phase-shifted in one direction and in the other region in the opposite direction, a so-called push-pull configuration.

The modulator in this study is fabricated by Polariton Technologies on a silicon photonics platform, allowing to efficiently route the optical signal while maintaining a compact footprint. The slot is 9 μm in length and 120 nm in width. This way, the active region requires a footprint of only ~5 μm². A microscope image of the device is shown in Fig. 4b. The narrow slot generates a high electric field under an applied voltage and, combined with the strong effective Pockels coefficient, results in a significant phase change. The large phase shift enables a highly compact design, ensuring that the modulator behaves as a lumped capacitance while achieving high electro-optical bandwidths. A broadband Pockels coefficient in the nonlinear material is essential to maintain this broadband performance. Within the modulator's bandwidth, and under null-point biasing, the optical field at the output is given by

$$E = j\alpha(L) \sin\left(\frac{V_{RF}}{V_{\pi L}^{(PS)}} L\pi\right) E_0 e^{j\omega t}, \tag{1}$$

where $E_0$ is the optical input field at frequency $\omega$, $\alpha(L)$ the total optical transmission (dependent on the plasmonic slot length $L$), $V_{\pi L}^{(PS)}$ the voltage-length-product of one of the phase shifters, and $V_{RF}$ the voltage across the plasmonic slot. The total optical transmission is composed of

$$\alpha(L) = \alpha_{Phot} \alpha_{CPL} e^{-\beta L}, \tag{2}$$

where $\alpha_{Phot}$ captures photonic losses (e.g., routing losses, grating coupler losses), $\alpha_{CPL}$ accounts for photonic-plasmonic-photonic coupling loss and $e^{-\beta L}$ is the plasmonic propagation loss. A sinusoidal RF field at frequency $\Omega_{RF}$ and with a power of $P_{RF}$ (measured into $R_{50} = 50\,\Omega$), yields a peak voltage across the slot of

$$V_{RF} = 2 \cdot \sqrt{2 P_{RF} R_{50}}. \tag{3}$$

Due to the capacitive nature of the load, the signal is reflected, doubling the voltage amplitude compared to the $R_{50}$ load. Applying the Jacobi-Anger expansion to Eq. (1), the fields of the upper (+) and lower (-) sideband read

$$E_{\pm} = j\alpha(L) J_1\left(\frac{V_{RF}}{V_{\pi L}^{(PS)}} L\pi\right) e^{j\phi_{RF}} E_0 e^{j(\omega \pm \Omega_{RF})t}, \tag{4}$$

where $J_1$ denotes the first order Bessel function of the first kind, and $\phi_{RF}$ the phase of the RF field. Note that the phase information of the RF fields is mapped to the phase of the optical field and the amplitude of the RF field translates to the optical amplitude through the Bessel function. This way, the transmitted complex symbols are mapped to complex symbols in the optical domain. Assuming small-signal conditions, i.e., linearizing $J_1$ around $V_{RF} = 0\,V$, the intensities of the upper and lower sideband $I_{\pm}$ become

$$I_{\pm} = |\alpha(L)|^2 \left(\frac{V_{RF}}{2 V_{\pi L}^{(PS)}} L\pi\right)^2 \cdot P_{opt}, \tag{5}$$

where $P_{opt} \propto |E_0|^2$ is the optical input power. Substituting Eq. (3) into Eq. (5), one obtains

$$I_{\pm} = |\alpha(L)|^2 \left(\frac{\sqrt{2 R_{50}}}{V_{\pi L}^{(PS)}} L\pi\right)^2 \cdot P_{opt} \cdot P_{RF}, \tag{6}$$

where the first term is the conversion efficiency $\eta$, i.e. $\eta := I_{\pm} P_{opt}^{-1} P_{RF}^{-1}$ with $[\eta] = W^{-1}$. This conversion efficiency measures the sideband

power for a given RF power $P_{RF}$ and optical power $P_{opt}$. The conversion efficiency depends on the cross section of the plasmonic waveguide and the nonlinear material through $V_{\pi L}^{(PS)}$ and $\beta$, and on the slot length through $|\alpha(L)|^2 L^2$. Taking the cross-section geometry, i.e., $V_{\pi L}^{(PS)}$ and $\beta$, as fixed, the slot length $L^*$ that maximizes the conversion efficiency is the characteristic decay length of a (surface) plasmon polariton, i.e.,

$$L^* = \frac{1}{\beta} \tag{7}$$

Further details can also be found in ref. 54. The conversion efficiency then reads

$$\eta = \frac{\pi^2}{e^2} R_{50} \cdot \left(\frac{|\alpha_{Phot} \alpha_{CPL}|}{V_{\pi L}^{(PS)} \beta}\right)^2. \tag{8}$$

From Eq. (8), it becomes clear that the conversion efficiency is large when the losses of the coupling and photonic parts are small and the $V_{\pi L}\beta$ of the plasmonic cross section is small. The $V_{\pi L}$ can be approximated by

$$V_{\pi L}^{(PS)} = \lambda \cdot \frac{1}{n_{mat}^3 r_{eff}} \cdot \frac{w_{slot}}{\Gamma}, \tag{9}$$

where $\lambda$ is the wavelength, $n_{mat}$ and $r_{eff}$ the refractive index and the effective Pockels coefficient of the nonlinear material, respectively, $w_{slot}$ is the slot width and $\Gamma$ is the interaction coefficient[55]. The material properties for organic materials can reach up to $n_{mat}^3 r_{eff} = 8700\,V^{-1}pm$[56]. The interaction coefficient $\Gamma$ takes into account how much the effective refractive index changes compared to the bulk material change upon applied voltages. For plasmonic waveguides, the interaction coefficient $\Gamma$ can be close to 1, as there is a good overlap between the RF and optical fields[55,57,58]. Furthermore, plasmonic waveguides can have a uniquely narrow slot width, yielding very low, in-device $V_{\pi L}^{(PS)}$ down to 150 Vμm[59]. While narrowing the slot reduces $V_{\pi L}^{(PS)}$, it increases plasmonic losses $\beta$. Interestingly, simulations show that the product $V_{\pi L}^{(PS)}\beta$ continues to decrease with narrower slots. However, excessively narrow slots degrade organic nonlinearity, making $w_{slot} \approx 100$ nm a practical optimum[60]. Assuming negligible photonic losses ($|\alpha_{Phot}|^2 = 1$) and coupling loss $|\alpha_{CPL}|^2 \wedge = 2 \cdot 0.5$ dB and $V_{\pi L}^{(PS)}\beta \wedge = 100$ VdB[58], the achievable conversion efficiency is around −35 dBm.

The above analysis holds within the modulator's RF bandwidth. For the modulator in this study, we measured a flat (within a standard deviation of 1.1 dB) frequency response up to 350 GHz as shown in Fig. 4e. This in good agreement with previous frequency response measurements, where flat responses up to 500 GHz and bandwidths of 880 GHz and 997 GHz have been shown[42,43]. The setup used to measure the frequency response is depicted in Fig. 4d. It comprises a tunable laser source emitting a carrier around 1550 nm. That carrier is then modulated by an RF source. The RF source provides a sinusoidal signal up to 70 GHz. To reach higher frequencies, mixers were used, see Methods section. Upon the sinusoidal RF signal, sidebands emerge, see Eq. (4), which were measured using an optical spectrum analyzer (OSA). Furthermore, with the OSA, the residual carrier was monitored to ensure the MZM is operated in the null point throughout the measurement. From the power in the sideband, which was measured by the OSA, the frequency response can be deduced[60,61]. We estimate that the total capacitance of the two plasmonic slots and contact pads is around 4 fF, as the electrical contact design is similar as in Horst et al.[43]. This leads to an energy consumption of 1 fJ/bit/V² – assuming four-level quadrature amplitude modulation (4QAM) – demonstrating the modulator's energy efficiency[62], and a bandwidth of ~800 GHz.

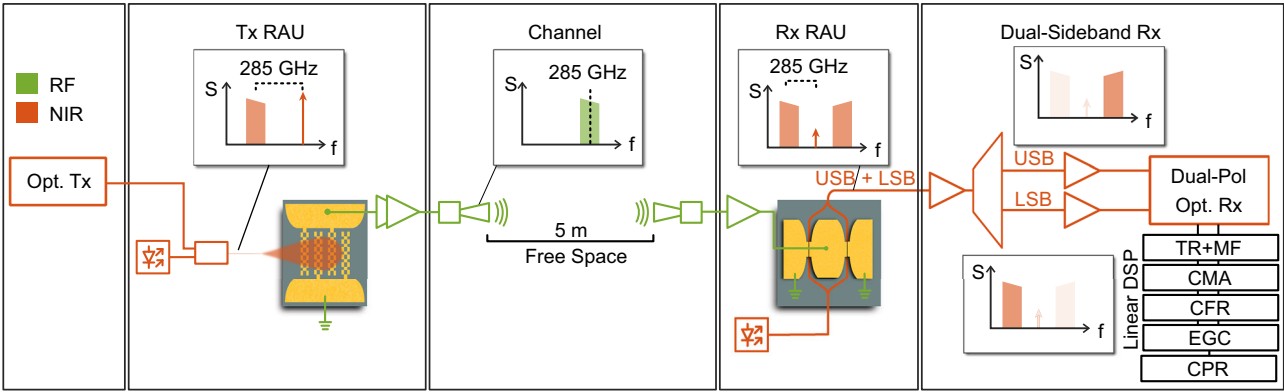

**Fig. 5 | Schematic of the experimental setup for all-plasmonic wireless data transmission connecting an optical transmitter to and optical receiver.** A 1550 nm optical signal is modulated via an IQ modulator, down-converted to 285 GHz at the transmitter remote antenna unit (Tx RAU) using a plasmonic photodetector, transmitted over a 5 m free-space link, and upconverted by a plasmonic MZM at the receiver remote antenna unit (Rx RAU). The recovered upper sideband (USB) and lower sideband (LSB) signals are processed in a dual-polarization optical receiver (Dual-Pol Opt. Rx) followed by linear digital signal processing (DSP) – timing recovery (TR), matched filtering (MF), constant modulus algorithm (CMA), carrier frequency recovery (CFR), equal gain combining (EGC) and carrier phase recovery (CPR). Radio frequency (RF) and near-infrared paths are color-coded in green and red, respectively.

Additionally, fiber-to-fiber loss is measured at 15.1 dB, including grating coupler losses (2 × 4.0 dB), plasmonic converter losses (2 × 1 dB), and plasmonic propagation losses (4.5 dB). The measured spectrum at applied DC voltages is shown in Fig. 4c. The spectrum follows an MZ interference curve as one MZM arm is slightly larger then the other, creating an imbalance. The peak-to-peak voltage required for switching the MZM from the minimum to the maximum transmission ($V_\pi^{(MZM)}$) is 11.1 V. However, due to the lumped-capacitor nature of the plasmonic modulator, see Eq. (3), the voltage induced by an incoming RF signal at a certain power is twice that of a conventional 50 Ω terminated modulator. This voltage enhancement leads to an effective $V_\pi^{(MZM)}$ of 5.5 V, highlighting the advantage of plasmonic modulators over their 50 Ω counterparts. With those values, the conversion efficiency, see Eq. (8), is −42.1 dBm.

## Sub-THz wireless data experiments

We now test the plasmonic photodetector and plasmonic modulator in a wireless data transmission experiment. For this purpose, we construct the setup illustrated in Fig. 5, which consists of an optical transmitter (Tx), a transmitting remote antenna unit (RAU) incorporating the plasmonic photodetector (PD), a 5 m free-space channel, a receiving RAU featuring the plasmonic modulator, and an optical receiver (Rx). See Methods Sections for details on the setup. This configuration enables data transmission from the optical transmitter to the optical receiver at a line rate of 120 Gbit/s over an all-plasmonic, single-polarization wireless channel operating at a 285 GHz carrier frequency.

At the optical transmitter, a 1550 nm carrier is modulated with a data signal using a 38 GHz IQ modulator. Digital Nyquist frequency division multiplexing (NFDM) partitions the data signal into six tributaries[63,64]. Each tributary carries symbols in a quadrature amplitude modulation (QAM) format at a rate of 8 GBd, resulting in a total transmission rate of 48 GBd over the channel. The optical signal is then routed to the transmitter RAU, where it is down-converted using a plasmonic graphene PD that beats the optical signal with a local oscillator (LO). The data signal and LO are free-space coupled into the active region of the PD. The bias and gate voltages are set to 0 V and −8 V, respectively. The LO frequency is offset by 285 GHz relative to the data carrier, generating an RF signal at 285 GHz. The PD outputs an RF power of −42 dBm in these experiments.

The 5 m free-space channel introduces a path loss of -96 dB, which is compensated using two horn antennas (2 × 30 dB) and two lenses (2 × 13.5 dB), reducing the residual effective channel loss to 9 dB. RF amplifiers further boost the signal, delivering 5.5 dBm RF power (measured into 50 Ω) at the plasmonic MZM modulator in the receiving RAU. The RF amplifier chain exhibits a bandwidth of around 30 GHz, which is the link's main bandwidth limitation. As mentioned, we project a significant improvement of the graphene PD, which will make the RF amplifiers obsolete. Thereby, the frequency bandwidth of the link could be drastically increased, and the capacity of the channel is further improved. The plasmonic MZM is optically fed by a C-band laser source via a grating coupler. The RF signal directly drives the MZM in a push-pull configuration without requiring an intermediate frequency. The modulator operates at its null point, imprinting the signal onto the optical amplitude. As a result, a suppressed carrier and two sidebands (LSB and USB), spaced by 285 GHz, emerge and are coupled out via a grating. The signal is then processed by a dual-sideband (DSB) receiver[65].

In the DSB receiver, the optical signal is amplified by an erbium-doped fiber amplifier (EDFA) and subsequently split into the USB and LSB signals with a wave shaper. These sidebands are detected using a conventional dual-polarization optical receiver comprising two LOs to mix the two sidebands down to basebands. Since both sidebands carry identical information but exhibit uncorrelated noise contributions, their combination provides an SNR advantage of up to 3 dB[65]. This combination is performed digitally in the signal processing (DSP) chain, which includes timing recovery (TR)[66], a matched filter (MF), a 37-tap constant modulus algorithm (CMA), carrier frequency recovery (CFR), an equal gain combiner (EGC), and carrier phase recovery (CPR). In the EGC, the LSB and USB signals are combined using a method adapted from[67]. Notably, we benefit from the linearity of the PD and MZM by employing linear equalizers, which reduces the required computational costs compared to nonlinear signal processing methods required in electronic receivers[68].

The frequency response, depicted as the black curve Fig. 6a, is obtained by beating lasers into the PD and measuring the optical power in one of the sidebands. The measured bandwidth is ~30 GHz, limited primarily by the RF amplifiers. Figure 6a also shows the power spectrum of the electrical signal corresponding to the LSB at the receiver, where all six tributaries are distinctly visible and color-coded accordingly. Notably, tributaries 2–4 exhibit higher power levels than the others due to the frequency response of the link. A suitable modulation format is assigned to each tributary via bit-loading, with the high-power tributaries 2–4 carrying 8QAM, while the remaining tributaries use a

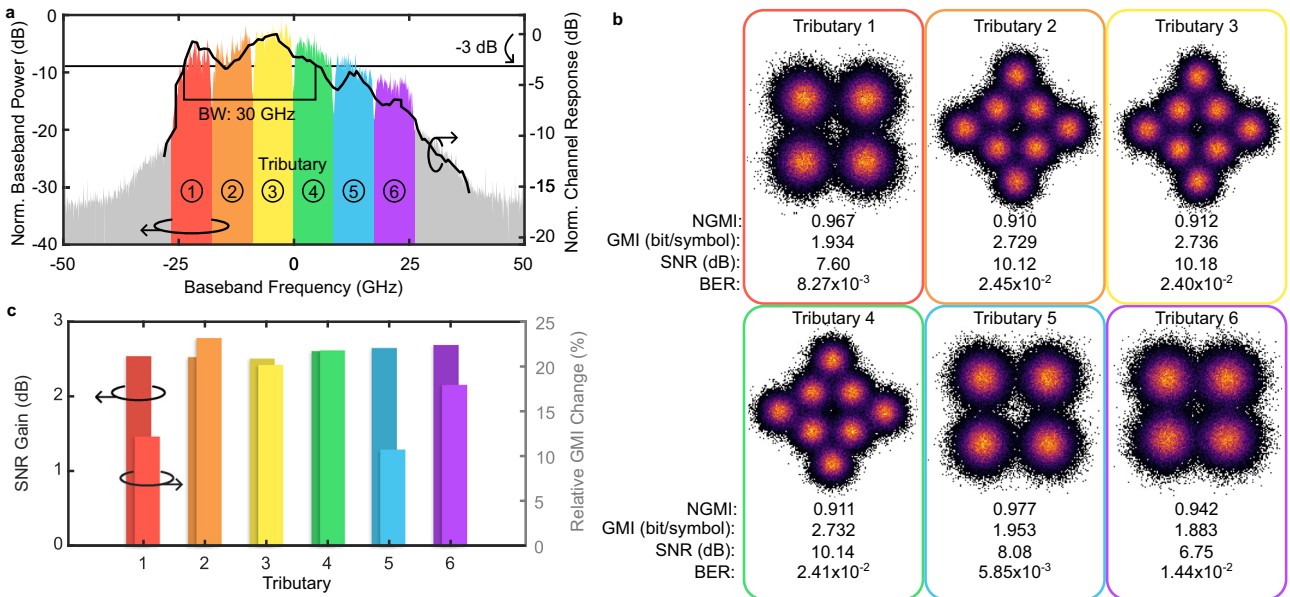

Fig. 6 | **Performance analysis of the all-plasmonic wireless transmission system.**
**a** Left axis shows the power spectrum of the received baseband signal, color-coded by tributary. Right axis shows the measured frequency response of the link having a bandwidth of 30 GHz. **b** Constellation diagrams of the six tributaries, with color-coded frames corresponding to their respective frequency bands. Below the normalized generalized mutual information (NGMI), the general mutual information (GMI), the signal-to-noise ratio (SNR) and bit error rate (BER) are provided. **c** Signal-to-noise ratio (SNR) gain (left/dark axis) and relative generalized mutual information (GMI) change (right/light axis) for each tributary when dual-sideband reception is used relative to single-sideband reception.

lower-order modulation format. Additionally, the relative power of each tributary is adjusted to further optimize the throughput[69].

Figure 6b presents the constellation diagrams for each tributary alongside key performance metrics: generalized mutual information (GMI), normalized GMI (NGMI), signal-to-noise ratio (SNR), and bit error rate (BER). For all tributaries, the NGMI exceeds the 0.9090 threshold required for soft-decision (SD) forward-error correction (FEC) concatenated with hard-decision (HD) FEC[70]. Moreover, the BER remains below the HD-FEC limit of $1.44 \times 10^{-2}$, allowing for the use of computationally less demanding HD-FEC. The HD-FEC limit of $1.44 \times 10^{-2}$ corresponds to a code rate of 0.8333[70]. The maximum achievable information rate is 111.7 Gbit/s, calculated by multiplying the average GMI of 2.328 bit/symbol with the total symbol rate of 48 GBd.

Figure 6c illustrates the performance gain achieved by using DSB reception instead of single-sideband reception. The plot shows the SNR gain (dark colors) and the relative GMI improvement (light colors) for each tributary. The SNR gain and GMI improvement are evaluated relative to the results where the EGC stage in the DSP is bypassed. The SNR gain, shown on the left axis, is ~2.5 dB. This translates to a GMI improvement of 10–20%, as shown on the right axis.

## Discussion

To conclude, we demonstrated a high-speed sub-THz wireless link exceeding 100 Gbit/s using plasmonic opto-electric and electrooptic converters. This marks the first successful use of a plasmonic graphene photodetector to generate the sub-THz signal in such a link. Both the plasmonic photodetector and modulator exhibit bandwidths beyond 350 GHz, enabling broadband, linear, and compact wireless-optical-wireless communication with unrestricted on-demand carrier frequency.

The link achieved a line rate of 120 Gbit/s at 285 GHz over a 5-m free-space distance, showcasing the viability of plasmonic components for high-frequency wireless links. Their flat frequency response, small footprint, and low fabrication costs eliminate key limitations of conventional III-V and electronic approaches while allowing direct integration with silicon photonics or other platforms. The devices' linearity enables the use of simple DSP, further reducing the energy consumption associated with the digital signal processing required in electronic approaches.

This work highlights the potential of plasmonic technology as a scalable, cost-effective platform for future high-speed networks. All-plasmonic links offer a clear path toward flexible and compact sub-THz transceivers, addressing the growing demand for data in 5G, 6G and beyond.

## Methods

### Plasmonic photodetector fabrication

Plasmonic photodetectors were fabricated on standard silicon substrates, with intrinsic doping and a specified resistivity of $10^4–10^6$ Ohm cm (MicroChemicals). The samples are thermally oxidized to form a 500-nm thick silicon dioxide layer. Electron beam lithography (EBL) and a lift-off process were used to define the lowest layer of the metamaterial and the gate pads. The 100-nm thick gold layer was deposited by electron beam evaporation. Next, the aluminum oxide spacer layer (120-nm thick) was grown in a plasma-enhanced atomic layer deposition (ALD) process. Commercially available bilayer graphene was transferred to the substrate by an external company (Graphene Platform). The transferred graphene was patterned with EBL and reactive ion etching. The metamaterial resonators were created in two steps with EBL, electron beam evaporation and lift-off process. Half the resonator and contact lines consist of 100 nm gold, whereas the other half deposited in the 2nd step consist of 7 nm silver and 93 nm gold. In the last step, the structure was encapsulated by a thermal ALD growth of 50 nm of alumina. Pads are opened by standard photolithography and wet chemical etching.

We used two devices on the same sample for the characterization and data measurements. A first device is used for Figs. 2 and 3, whereas a second device is used in the experiments in Figs. 5 and 6. The two devices have the same dimensions. The dimensions of the devices used are as follows: The metamaterial consists of 6 by 10 unit cells, where each unit cell has a size of 1170 nm by 700 nm. Trough the center of the unit cells the interdigitated electrode line is positioned which has a width of 90 nm. In the center, perpendicular to the interdigitate electrode, the plasmonic resonator is positioned which has a length of 246 nm and a width of 90 nm. A visual representation is provided in the Supplementary Fig. 2.

## Plasmonic photodetector design

As outlined in the main text, the metamaterial design follows a similar architecture as in ref. 47. To achieve the factor ×4 improvement in responsivity and the factor ×12 higher output photocurrent, the following design considerations were taken into consideration.

The plasmonic graphene PD in operation is modeled as an equivalent circuit consisting of

- The resistance of the interdigitated electrodes including the resonators
- The resistance of the graphene channel between two interdigitated electrodes
- The resistance of the contact between the interdigitated electrodes and the graphene
- The current source representing the photocurrent
- The load resistance to which the PD delivers the current.

These values per unit cell are calculated from the physical parameters of the metamaterial structure, i.e., unit cell size, interdigitated electrode width and height. The parameters were optimized in a multidimensional optimization scheme where the optical absorption given by the metamaterial, the RF power delivered to the load resistance and the optical power distribution were considered. We note that the electrical design of the metamaterial cannot be arbitrarily changed without influences the optical resonant behavior of the metamaterial. Therefore, such a multi-dimensional optimization approach is required. Theoretically, responsivities of 100 mA/W are possible as outlined in ref. 47. For example, improving the graphene quality is one factor that can improve the responsivity. This was in part emulated by cooling the device to cryogenic temperatures, where >4× factor responsivity increase was achieved[71].

## Details on experimental environment

All characterization and wireless data transmission measurements were performed in the same laboratory environment. The laboratory is climate controlled to 22 °C +/− 1 °C with a relative humidity centered around 50% +/− 5%.

## Low-frequency photodetector characterization

IV and gate curve measurements were performed using a precision source measure unit directly in the same optical setup as described in the main text and Fig. 2e.

Low-frequency measurements with the lock-in amplifier were performed by using a similar setup as presented in Fig. 2e. A tunable laser source is fed to an erbium doped fiber-amplifier (EDFA) to boost the optical input power. This signal is modulated with an acousto-optic modulator at a frequency of 320 kHz. The optical signal is then attenuated with a variable optical attenuator (VOA). In a last stage, the polarization is controlled by a manual polarization rotator and fed to the device through the single mode optical fiber which is controlled on a piezo 3-axis stage (SmarAct). The electrical signal is again read out through the RF probe, and the AC component is fed through the bias tee to the lock-in amplifier (Zürich instruments).

## High-frequency photodetector characterization

The RF frequency response of the plasmonic photodetectors was performed as outlined in the main text. The device was connected to the ESA (Keysight UXA N9041B) with high-frequency GSG probes, a bias tee, and a 2.4 mm RF cable. The RF losses of the probes are compensated with the measurement report provided by the vendor. The bias tee and cable losses are compensated through reference measurements with a commercial high-frequency photodetector. Measurements in the 110–170 GHz and 220–330 GHz range employ sub-harmonic mixers (Virginia Diodes, SAX modules). Mixer conversion losses and probe losses are removed through the respective calibration data provided by the vendors.

## Frequency modulator characterization

The static $V_\pi$ of the plasmonic modulator was measured by applying a static voltage across the modulator. Due to an imbalance in the length of the two Si waveguides forming the MZM, the intensity at the output depends on the wavelength. An applied voltage leads to a shift of that intensity spectrum from which the $V_\pi$ is estimated, see Fig. 4c. Concretely, we extracted the free spectral range (FSR), i.e. 6.34 nm, and the wavelength shift upon applying a voltage of ±1 V. Within a linearization, the FSR corresponds to a phase shift of π. This is then used to estimate the phase shift induced by applying ±1 V, i.e. 0.56 nm. Linearly extrapolating the required voltage to induce a π-shift reveals the $V_\pi$. Further details can be found in ref. 59. Higher frequencies were measured using the setup in Fig. 4d. Thereby, we set the wavelength of the laser source around 1550 nm such that the MZM is operated in the null-point. The sinusoidal RF signal is generated using an RF signal generator (Keysight E8257D) up to 70 GHz. The frequency region between 70 and 110 GHz was generated using the RF signal generator with the multiplier (Radiometer Physics AFM6). The frequency region between 110 and 350 GHz was generated using VNA extenders (Virginia Diodes WR 6.5, WR 5.1, WR 4.3 and WR 3.4). For small frequencies, a high-resolution OSA (Apex AP2040) was used; for larger frequencies, the Yokogawa AQ6370C was used. The frequency response was measured at logarithmically spaced points, with a step size of ~5 GHz at the higher end. Finer spacing was avoided, as the OSA's resolution bandwidth is 4 GHz and the frequency response follows a simple RC-response, see Horst et al.[43]. For calibration of the electric driving circuit, we used sub-harmonic mixers (Virginia Diodes, SAX modules) as discussed above. The remaining mixer conversion losses and probe losses are calibrated using the data provided by the vendor.

## Data transmission setup

In this section, we provide details on each part of the data transmission setup:

The optical transmitter consists of: - a laser source (NKT), a 38 GHz IQ modulator (Oclaro), which is driven by a 256 GSa/s arbitrary waveform generator (Keysight) - a tunable laser source (Keysight) functioning as local oscillator - an erbium-doped fiber amplifier (Keopsys)

The sub-THz emitter consists of: - the plasmonic PD as illustrated in Fig. 2e where DC control and read-out is done by a source-measure unit (Keysight) - the plasmonic PD is contacted by a high-frequency probe for operation from 225 to 330 GHz (GGB Industries inc.) - the sub-THz signal is amplified by a low-noise amplifier (Radiometer Physics GmbH) and a medium power amplifier (VDI) - the sub-THz signal is then radiated into free-space with a horn antenna (Flann) and collimated with a Teflon lens.

The sub-THz receiver consists of: - the plasmonic modulator where light from a tunable laser source (Keysight) is coupled by a fiber array to the device - the plasmonic modulator is contacted by a high-frequency probe for operation from 225 to 330 GHz (GGB Industries inc.) - the sub-THz signal is coupled to the probe by a Teflon lens, a horn antenna (Flann) and amplified by a medium power amplifier (Frauenhofer)

The optical receiver consists of: - the dual sideband receiver setup discussed in detail in ref. 65.

## Data availability

Data from this work is provided in the manuscript, supplementary information figures and table and from the corresponding author upon request.

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

## Acknowledgements

This work was supported by the Hans-Eggenberger Stiftung, the European Union's Horizon 2020 Research and Innovation Program through the project aCryComm, FET Open Grant Agreement no. 899558, HORIZON.2.4 projects ECO-eNET (101139133), Proteus (101139134), Allegro (101092766), by the Swiss State Secretariat for Education, Research, by the Innovation (SERI) through the SwissChips research project, by the Adrian Weiss Stiftung/ETH Grant (22-2 ETH-037) and the EMPIR programme project SuperQuant (20FUN07), which has received funding from the EMPIR program co-financed by the Participating States and from the European Union's Horizon 2020 research and innovation program. The Swiss National Science Foundation (SNF) is acknowledged for support through the REquip program (206021_198113). We thank the Cleanroom and Operations team of the Binning and Rohrer Nanotechnology Center (BRNC) for their support. Polariton Technologies thanks Lightwave Logic for supplying the Perkinamine™ chromophore series 3 electro-optic material.

## Author contributions

T.B., S.M.K., Y.H., J.S. and J.L. conceived the overall idea and led the development of the system experiment. T.B. designed the plasmonic modulator which was provided by Polariton Technologies, represented by M.D., T.B. and Y.H. measured the modulator's device characteristics. S.M.K., A.Z. and T.B. developed the photodetector structure, where establishing the concept and structure was supported by J.L., S.M.K. fabricated the devices with support from Y.F., S.M.K., D.R. and M.B. established the characterization protocol for the photodetectors and performed the characterization measurements. T.B., A.Z and S.H. conducted the data experiment and were supported by L.K., T.B. and S.M.K. jointly prepared the manuscript, which was revised by all co-authors.

## Funding

## Competing interests

The authors declare no competing interests.
