## [Transparent Peer Review file · Nature Communications]

All-Plasmonic Sub-Terahertz Wireless Communication Link

Corresponding Author: Mr Tobias Blatter

Version 0:

Reviewer comments:

Reviewer #1

(Remarks to the Author)

The authors first all-plasmonic wireless communications using photodetectors and modulators in 0.1-0.3 THz. A lab experiment validates that the proposed communication system achieved 120 Gbit/s over a 5-meter transmission distance. The authors highlight that the potential of all-plasmonic technology for future high-speed, low-power, and cost-effective wireless-optical-wireless networks. However, there are many confusing aspects of this manuscript, so I cannot give a positive evaluation at this time. Here are my comments:

1. The novelty of this manuscript is ambiguous. The authors assert that no prior research has been conducted on plasmonic graphene photodetectors (PDs), but in reality, numerous papers related to plasmonic graphene PDs have already been published, as referenced in [R1]-[R4]. Although the operating frequencies of these prior works may not all fall within the sub-terahertz range, it is relatively straightforward to adjust the frequency based on their underlying principles.
2. Another key innovation the authors aim to highlight is the broadband capability (0.1-0.3 THz). However, other existing technical solutions appear to offer significantly wider bandwidths, as indicated in [46]. Therefore, it is unclear what specific advantages this work possesses and what its potential applications might be. Additionally, in comparison to the graphene photodetector described in [46], it would be beneficial to elucidate the differences, such as in fabrication technology or material characteristics.
3. Figure 4c only illustrates the relationship between wavelength and transmission parameters when a bias voltage of $\pm 1V$ is applied. It would be more informative to display the transmission parameters at other voltage levels, which could potentially enable a $180^\circ/360^\circ$ phase shift.
4. There is a typographical error in line 216. The phrase "Figure 4Figure5" should be corrected to "Figure 5." Please review the manuscript for accuracy.
5. At sub-terahertz and higher frequencies, the propagation loss tends to be more substantial. Consequently, for the proposed plasmonic photodetector, it would be valuable to explore whether its responsivity can be further enhanced and to provide specific methods for achieving this improvement.
6. In the context of communication systems, the authors should provide more detailed information, such as the structure of the antennas and the specific experimental setup, to better support their claims.
7. If the authors want to prove that the method proposed in this paper has better cost advantages, sufficient literature comparisons should be provided, including other performance indicators, to highlight the advantages of the paper.
8. The amount of data in Figure 4e is a bit small. It is recommended that the authors make more measurements to provide the average value and error range.

[R1] Ding, Y., Cheng, Z., Zhu, X., Yvind, K., Dong, J., Galili, M., Hu, H., Mortensen, N., Xiao, S. & Oxenløwe, L. (2020). Ultra-compact integrated graphene plasmonic photodetector with bandwidth above 110 GHz. *Nanophotonics*, 9(2), 317-325. <https://doi.org/10.1515/nanoph-2019-0167>

[R2] Chen, Z. et al. Synergistic Effects of Plasmonics and Electron Trapping in Graphene Short-Wave Infrared Photodetectors with Ultrahigh Responsivity. *ACS Nano* 11, 430-437 (2017). <https://doi.org/10.1021/acsnano.6b06172>

[R3] Gosciniak, J., Rasras, M. & Khurgin, J. B. Ultrafast Plasmonic Graphene Photodetector Based on the Channel Photothermoelectric Effect. *ACS Photonics* 7, 488-498 (2020). <https://doi.org/10.1021/acsp Photonics.9b01585>

[R4] Cai, H., Yang, C., Shen, L., Yu, Y. & Zhang, X. High-Efficiency and Polarization-Independent Waveguide-Integrated Graphene Plasmonic Photodetectors Operating at 2 μm . *ACS Photonics* 11, 1565-1573 (2024). <https://doi.org/10.1021/acsp Photonics.3c01755>

Reviewer #2

(Remarks to the Author)

Reviewer #3

(Remarks to the Author)

The manuscript introduces an all-plasmonic sub-THz wireless link, which enables flexible modulation of the carrier frequency. It also verifies that the previously proposed plasma devices could function as receivers and transmitters, facilitating the conversion of optical and broadband-stabilized terahertz signals. Finally, the authors experimentally demonstrate information transmission over a spatial distance of 5 meters at 285 GHz with data rates of up to 120 Gbit/s. This achievement shows the potential for seamless connection between optical fibers and THz components in short-range applications. While the manuscript provides experimental validation, it lacks sufficient theoretical support.

Furthermore, the devices and methods used in the study are based on previously published work and do not present significant innovations. Therefore, I do not recommend it for publication in a high-level journal such as Nature Communications. The following are some additional comments:

1. The use of "However" in line 77 appears incorrect. Additionally, the manuscript does not explain why graphene photodetectors (PDs) in the sub-terahertz band have not been previously studied. This omission leaves a gap in the narrative that needs to be addressed.
2. Figures 1a and 1b lack necessary scale labeling, making it difficult to identify the specific data measurement area discussed in the METHODS section. Moreover, the manuscript would benefit from providing more detailed parameters and conditions of the laboratory environment, such as humidity and temperature, to ensure reproducibility and clarity.
3. Line 125 references supplementary information, but it appears that this material has not been uploaded.
4. Line 184 mentions an interaction between the RF field and the optical field. Could a more in-depth study and elaboration be given?
5. Most importantly, as stated in the paper, the plasma devices used have all been published, so are there any different effects or innovations in the devices used in this paper? Moreover, it is suggested that some corresponding physical mechanisms or details be added to this paper as well to improve the readability and novelty of the article.
6. In line 216, it should be figure5, not figure4figure5.
7. The authors claim superior performance for their work. To substantiate these claims, it is suggested that a detailed table comparing key metric parameters with existing literature be included. This would provide a clearer benchmark for evaluating the manuscript's contributions.
8. Several details in the text need to be reviewed for greater rigor, as previously highlighted in points 1, 3, and 6.

Version 1:

Reviewer comments:

Reviewer #1

(Remarks to the Author)

The authors have addressed my questions. I am satisfied with the revisions and agree to the manuscript being published in Nature Communications

Reviewer #2

(Remarks to the Author)

Reviewer #3

(Remarks to the Author)

The authors have addressed all my concerns in the review.

Eidgenössische Technische Hochschule Zürich
Swiss Federal Institute of Technology Zurich

Institute of Electromagnetic Fields (IEF)

ETH Zurich
Tobias Blatter | Dr. Sc. Stefan M. Koepfli
PhD Candidate | Postdoctoral Researcher
ETZ K77 | ETZ K60.1
Gloriastrasse 35
8092 Zurich, Switzerland

Email : blattert@ethz.ch
Email: koepflis@ethz.ch
Web: <http://www.ief.ee.ethz.ch/>

Zurich, 8 July 2025

Revisions on *Nature Communications Manuscript NCOMMS-25-24299*

Dear Reviewers,

We extend our appreciation to the reviewers for their valuable comments and comprehensive review, aiding in the enhancement of our paper's quality. All comments from the reviewers have been thoroughly addressed, and corresponding changes have been made to the manuscript. Below are detailed responses to each of the reviewers' comments, highlighted in **blue** for clarity. Additionally, alterations made to the initial manuscript are clearly marked in **red** below each comment and in the updated version of the paper.

Yours sincerely,

Tobias Blatter and Stefan M. Koepfli

Reviewer 1

“The authors first all-plasmonic wireless communications using photodetectors and modulators in 0.1-0.3 THz. A lab experiment validates that the proposed communication system achieved 120 Gbit/s over a 5-meter transmission distance. The authors highlight that the potential of all-plasmonic technology for future high-speed, low-power, and cost-effective wireless-optical-wireless networks. However, there are many confusing aspects of this manuscript, so I cannot give a positive evaluation at this time. Here are my comments.”

We thank the reviewer for his overall feedback. We regret that have been confusing aspects in our text. We provide detailed comments to the reviewer’s comments below to address all the issues.

1. The novelty of this manuscript is ambiguous. The authors assert that no prior research has been conducted on plasmonic graphene photodetectors (PDs), but in reality, numerous papers related to plasmonic graphene PDs have already been published, as referenced in [R1]-[R4]. Although the operating frequencies of these prior works may not all fall within the sub-terahertz range, it is relatively straightforward to adjust the frequency based on their underlying principles.

[R1] Ding, Y., Cheng, Z., Zhu, X., Yvind, K., Dong, J., Galili, M., Hu, H., Mortensen, N., Xiao, S. & Oxenløwe, L. (2020). Ultra-compact integrated graphene plasmonic photodetector with bandwidth above 110 GHz. *Nanophotonics*, 9(2), 317-325. <https://doi.org/10.1515/nanoph-2019-0167>

[R2] Chen, Z. et al. Synergistic Effects of Plasmonics and Electron Trapping in Graphene Short-Wave Infrared Photodetectors with Ultrahigh Responsivity. *ACS Nano* 11, 430-437 (2017). <https://doi.org/10.1021/acsnano.6b06172>

[R3] Gosciniaik, J., Rasras, M. & Khurgin, J. B. Ultrafast Plasmonic Graphene Photodetector Based on the Channel Photothermoelectric Effect. *ACS Photonics* 7, 488-498 (2020). <https://doi.org/10.1021/acsp Photonics.9b01585>

[R4] Cai, H., Yang, C., Shen, L., Yu, Y. & Zhang, X. High-Efficiency and Polarization-Independent Waveguide-Integrated Graphene Plasmonic Photodetectors Operating at 2 μm . *ACS Photonics* 11, 1565-1573 (2024). <https://doi.org/10.1021/acsp Photonics.3c01755>

We thank the reviewer for his comment on the novelty of the manuscript. We regret that it was not sufficiently clear from our text what the new insights are. We did not claim that no prior research has been conducted on plasmonic graphene photodetectors. We state that plasmonic graphene photodetectors have not been tested in a sub-THz wireless communication scenario. We clarify this point by making the following challenges to the manuscript:

L76: ~~However~~, Recently, plasmonic photodetectors (PDs) based on graphene have been shown to reach bandwidths beyond 400 GHz⁴⁶ and even 500 GHz⁴⁷. ~~With this, the OE side is matching the speed of the EO side and could enable sub-THz wireless communication. However, plasmonic graphene PDs have not been explored or verified for sub-THz communication [...].~~

I319: To conclude, we demonstrated a high-speed sub-THz wireless link exceeding 100 Gbit/s using plasmonic opto-electric and electro-optic converters. This marks the first successful use of a plasmonic graphene photodetector to generate the ~~sub-THz~~ signal in such a link.

We comment on the four references provided by the reviewer:

[R1] is indeed a fast plasmonic graphene PD with high responsivity. No characterization beyond 110 GHz has been performed, so it is hard to make claims about the potential devices' bandwidth. However, we note that the highest reported output current for their device is 144 μA , which is for the longest device that already shows a drop in frequency response towards 110 GHz, so it is unlikely that operation at 300 GHz is possible. The shorter device that is faster does not reach 100 μA photocurrent. Our devices output more than 500 μA which is required to generate enough power.

[R2] reports on a plasmonic graphene PD which uses a gain assisted detection mechanism and its bandwidth is < 1 MHz. The device will not operate in the sub-THz.

	[R3] is pure simulation work without experimental verification. The claimed bandwidth of >500 GHz for a device operating in the PTE mode is most likely not achievable, see also Yoshioka, K., Wakamura, T., Hashisaka, M. et al. Ultrafast intrinsic optical-to-electrical conversion dynamics in a graphene photodetector. Nat. Photon. 16, 718–723 (2022). https://doi.org/10.1038/s41566-022-01058-z [R4] shows a plasmonic graphene PD with >14 GHz bandwidth. The VNA measurements and modelling predict a bandwidth of 185 GHz. Additionally, the highest reported photocurrent is <20 μA. To summarize: none of the reviewer's cited plasmonic graphene PDs have sufficient bandwidth while simultaneously offering a high enough output current; something that is generally true for plasmonic graphene PDs. We clarify that these are required properties by making the following changes to the main text: L79: However, plasmonic graphene PDs have not been explored or verified for sub-THz communication due to a lack of simultaneously achieving sufficient output powers and operation in the 100s GHz.
	2. Another key innovation the authors aim to highlight is the broadband capability (0.1-0.3 THz). However, other existing technical solutions appear to offer significantly wider bandwidths, as indicated in [46]. Therefore, it is unclear what specific advantages this work possesses and what its potential applications might be. Additionally, in comparison to the graphene photodetector described in [46], it would be beneficial to elucidate the differences, such as in fabrication technology or material characteristics.
	We thank the reviewer for pointing out that the claims on the broadband capability are not clear. The reviewer points out correctly that the device presented in our ref [46] have been demonstrated for >500 GHz bandwidth. In ref [46] Table S2 also outlines that there are only few technologies that are able to operate above 100 GHz. The device presented in this work has a similar architecture to the one presented in [46], however, the measurement setup at the moment only allows measurements up to 330 GHz. We make the following changes to the main text to clarify this characteristic: I139: The measurement shows no roll-off behavior in the measurement range verifying that the PDs can operate across a broad range and are able to generate the targeted sub-THz carriers. The here demonstrated > 330 GHz bandwidth is limited by the characterization setup and >500 GHz could be possible as demonstrated in [46], which is based on a similar architecture. With respect to the 2nd part of the reviewer's comment on the specific advantages and potential applications as well as the differences to ref [46]:  - The device presented here has a responsivity of 6 mA/W which is a factor x4 higher than the device in [46]. - The device is able to operate at higher optical input powers which, in combination with the higher responsivity allows to generate a higher photocurrent. The 0.556 mA reported here are a factor x12 higher than the first generation device in [46]. - These improvements allow for the specific application that we present in this work: wireless communication in the sub-THz at a carrier frequency at 280 GHz. However, due to the flat frequency response the device can also generate:  o RF signals in other bands as illustrated in Fig. 2f and Fig. 3b. o Drastically improve the bandwidth of sub-THz links. o Be used for classical optical communication. o Or any other application that requires an infrared photodetector.

We clarify these points by making the following additions to the main text:

L164: The maximum extracted photocurrent is 0.556 mA, which, to the best of our knowledge, is the highest photocurrent reported for any high-speed zero-bias graphene photodetector **as summarized in Figure 3c**. The changes which lead to the improvements in responsivity and maximum photocurrent compared to the prior work are further discussed in the Method section. With the here reported values the device can be used in any scenario requiring high-speed infrared photodetectors, for example data communication. Next, we will verify the usability of the photodetector as a sub-THz source in a wireless data transmission scenario with a carrier frequency of 285 GHz in the following sections after discussing the sub-THz receiver in the form of a plasmonic modulator.

Figure 1: DC photocurrent and RF power generation of plasmonic graphene photodetectors.

a) Generated DC photocurrent as function of optical power on the device. The slope leads to a responsivity of 6 mA/W. Gray dots correspond to measurements performed with a lock-in amplifier. Blue to green squares correspond to measurements where the PD was connected to the electrical spectrum analyzer and sub-harmonic mixers. b) Generated RF power as function of DC photocurrent for three RF frequencies. 70 GHz: blue, 140 GHz blue-green, 280 GHz green. c) Comparison of reported high-speed graphene photodetectors as function of achieved bandwidth and maximum output photocurrent. Detailed parameters for each data point can be found in Supplementary Table 1.

METHODS:

Plasmonic photodetector design

As outlined in the main text, the metamaterial design follows a similar architecture as in [46]. To achieve the factor x4 improvement in responsivity and the factor x12 higher output photocurrent, the following design considerations were taken into consideration.

The plasmonic graphene PD in operation is modelled as an equivalent circuit consisting of

- The resistance of the interdigitated electrodes including the resonators
- The resistance of the graphene channel between two interdigitated electrodes
- The resistance of the contact between the interdigitated electrodes and the graphene
- The current source representing the photocurrent
- The load resistance to which the PD delivers the current.

These values per unit cell are calculated from the physical parameters of the metamaterial structure, i.e., unit cell size, interdigitated electrode width and height. The parameters were optimized in a multi-dimensional optimization scheme where the optical absorption given by the metamaterial, the RF power delivered to the load resistance and the optical power distribution were considered. We note that the electrical design of the metamaterial cannot be arbitrarily changed without influences the optical resonant behavior of the metamaterial. Therefore, such a multi-dimensional optimization approach is required. [...]

L275: The RF amplifier chain exhibits a bandwidth of around 30 GHz, which is the link's main bandwidth limitation. As mentioned, we project a significant improvement of the graphene PD, which will make the RF amplifiers obsolete. Thereby, the frequency bandwidth of the link could be drastically increased, and the capacity of the channel is further improved.

We further added a Supplementary Table 1 that offers a detailed comparison to the current high-speed graphene photodetector state-of-the-art:

Ref.	Year	Bandwidth	Max. Output	Device Resistance	Det. Mech.	Responsivity	Bias Voltage	Gate Voltage	
1	2009	>40 GHz	---	a	~2000 Ω	PV	0.5 mA/W	0 V	80 V
2	2010	16 GHz	~63 μ A	b	140 Ω	PV	1.5 mA/W	0 V	-15 V
						-	6.1 mA/W	0.4 V	-15 V
3	2013	>20 GHz	~19.2 μ A	c	---	PV	15.7 mA/W	0 V	No Gate
4	2013	18 GHz	~2.6 μ A	-	---	PV	50 mA/W	0 V	No Gate
5	2014	3 GHz	~240 μ A	-	~100 Ω	PC	57 mA/W	0.4 V	9 V
6	2014	41 GHz	~20 μ A	-	187 Ω	PV	16 mA/W	0 V	No Gate
7	2015	42 GHz	~0.2 μ A	b	98 Ω	PTE	78 mA/W	0 V	2.9 V
8	2016	65 GHz	~1 mV ~15 μ A	a	254 Ω	PTE	3.5 V/W 35 mA/W	0 V	~6 V, ~6 V
						---	76 mA/W	0.3 V	~6 V, ~6V
9	2017	76 GHz	15.9 μ A	a	130 Ω	BOL	1 mA/W	1 V	No Gate
10	2018	>18 GHz	1.5 mV 7.5 μ A	x	200 Ω	PTE	4.7 V/W 48 mA/W	0 V	4 V, -2 V
						PC	170 mA/W	0.4 V	-8 V, 5 V
11	2018	> 50 GHz	3.7 μ A	c	68 Ω	PC	1050 mA/W	0.02 V	-20 V
12	2018	>110 GHz	274 μ A	b	100 Ω	BOL	400 mA/W	-0.6 V	No Gate
						PTE/PV	3 mA/W	0 V	No Gate
13	2018	38 GHz	~5.5 μ A	-	300 Ω	---	0.57 mA/W	0 V	No Gate
14	2019	42 GHz	2.9 mV ~1.83 μ A	x	~2000 Ω	PTE	12.2 V/W	0 V	-4 V, -6 V
15	2019	>110 GHz	144 μ A	d	100 Ω	PV/PC	360 mA/W	2.2 V	No Gate
16	2019	>70 GHz	15 μ A	a	---	PC	300 mA/W	0.5 V	No Gate
17	2020	>67 GHz	4 mV 8 μ A	a	~500 Ω	PTE	6 V/W 12 mA/W	0 V	3 V, -1 V
18	2020	>40 GHz	~87 μ A	b	~150 Ω	BOL	396 mA/W	0.3 V to 1 V	0 to 2.8 V
19	2021	70 GHz	15.1 mV ~214 μ A	b, x	~70 Ω	PTE	3.5 V/W	0 V	-1 V, -8 V
20	2021	70 GHz	208 μ A	a	---	PC	104 mA/W	0.8 V	No Gate
21	2021	>67 GHz	---	a	172 Ω	PC	100 mA/W	0.3 V	No Gate
22	2021	12 GHz	38.6 mV 32.2 μ A	b, x	~1200 Ω	PTE	90 V/W 75 mA/W	0 V	-0.5 V, -2.1 V
23	2022	78 GHz	233 μ A	b	231 Ω	BOL	603.9 mA/W	1.9 V	No Gate
24	2023	30 GHz	0.21 μ A	-	355 Ω	IPE/PTFE	0.33 mA/W	6 V	No Gate
25	2023	>500 GHz	45.8 μ A	-	235 Ω	PV	1.57 mA/W	0 V	-2.5 V
26	2024	36 GHz	142 μ A	b	150 Ω	PC	460 mA/W	0.5 V	No Gate
27	2024	420 GHz	18.5 mV 5.4 μ A	b	5806 Ω	PTE	1.6 V/W 0.4 mA/W	0 V	0.5 V
28	2025	>14 GHz	17 μ A	b	286 Ω	BOL	122 mA/W	2.6 V	No Gate
29	2025	155 GHz	---	a	255 Ω	PV/PTE/PC	68 mA/W	0.3 V	-38 V
This work	2025	>330 GHz	556 μ A	-	22 Ω	PV	6 mA/W	0 V	-6.75 V

	3. Figure 4c only illustrates the relationship between wavelength and transmission parameters when a bias voltage of $\pm 1V$ is applied. It would be more informative to display the transmission parameters at other voltage levels, which could potentially enable a $180^\circ/360^\circ$ phase shift.
	We thank the reviewer for this suggestion. We agree that a larger voltage directly reveals the V_{π}. However, the plasmonic-organic modulator is rated to operate at DC levels of about $\pm 1V$. Therefore, we can not provide the requested plot as operation at higher voltages could damage the device. Yet, we revised the manuscript by giving more details on how the V_{π} is calculated.  METHODS: Frequency modulator characterization The static V_{π} of the plasmonic modulator was measured by applying a static voltage across the modulator. Due to an imbalance in the length of the two Si waveguides forming the MZM, the intensity at the output depends on the wavelength. An applied voltage leads to a shift of that intensity spectrum from which the V_{π} is estimated, see Figure 4c. Concretely, we extracted the free spectral range (FSR), i.e. 6.34 nm, and the wavelength shift upon applying a voltage of $\pm 1V$. Within a linearization, the FSR corresponds to a phase shift of π. This is then used to estimate the phase shift induced by applying $\pm 1V$, i.e. 0.56 nm. Linearly extrapolating the required voltage to induce a π-shift reveals the V_{π}. Further details can be found in Reference ⁶¹.
	4. There is a typographical error in line 216. The phrase “Figure 4Figure5” should be corrected to “Figure 5.” Please review the manuscript for accuracy.
	We thank the reviewer for pointing out this typo. It has been corrected in the revised manuscript.
	5. At sub-terahertz and higher frequencies, the propagation loss tends to be more substantial. Consequently, for the proposed plasmonic photodetector, it would be valuable to explore whether its responsivity can be further enhanced and to provide specific methods for achieving this improvement.
	We thank the reviewer for the comment on the propagation losses at high frequencies. Indeed, increasing the responsivity or the RF output power of the PD could increase the distance, data rate and resilience to propagation losses in the sub-THz. A potential route to increase the responsivity is already outlined in our previous work, ref [46]. We note that the broadband characteristic of the PD also potentially allows one to switch the carrier frequency to different frequencies where the propagation losses depending on the weather conditions could be lower. To illustrate this, we performed the measurement at 70, 140, and 280 GHz as shown in Fig. 3b. We make the following changes to the main text to address the reviewer’s comment:  I154: Furthermore, the output RF power remains the same across the three tested frequencies (70, 140, and 280 GHz), showcasing that the flat frequency response remains even when operating the device at higher optical input powers. This means the PD can flexibly generate various carrier frequencies on demand, enabling dynamic adaptation to system and channel conditions. For instance, under fog or snowfall, lower carrier frequencies benefit from reduced atmospheric absorption ⁵².  I372: Theoretically, responsivities of 100 mA/W are possible as outlined in ⁴⁷. For example, improving the graphene quality is one factor that can improve the responsivity. This was in part emulated by cooling the device to cryogenic temperatures, where >4x factor responsivity increase was achieved ⁷².

	6. In the context of communication systems, the authors should provide more detailed information, such as the structure of the antennas and the specific experimental setup, to better support their claims.
	We appreciate the reviewer's feedback on the missing details on the experimental setup. We added the following changes to the main text to address this:  I257: For this purpose, we construct the setup illustrated in Figure 5, [...]. See Methods Sections for details on the setup.  I420: Data transmission setup In this section we provide details on each part of the data transmission setup: The optical transmitter consists of:  - a laser sources (NKT), a 38 GHz IQ modulator (Oclaro), which is driven by a 256 GSa/s arbitrary waveform generator (Keysight) - a tunable laser source (Keysight) functioning as local oscillator - an erbium-doped fiber amplifier (Keopsys) The sub-THz emitter consists of:  - the plasmonic PD as illustrated in Fig. 2a,b where DC control and read-out is done by a source-measure unit (Keysight) - the plasmonic PD is contacted by a high frequency probe for operation from 225 to 330 GHz (GGB Industries inc.) - the sub-THz signal is amplified by a low noise amplifier (Radiometer Physics GmbH) and a medium power amplifier (VDI) - the sub-THz signal is then radiated into free-space with a horn antenna (Flann) and collimated with a Teflon lens. The sub-THz receiver consists of:  - the plasmonic modulator where light from a tunable laser source (Keysight) is coupled by a fiber array to the device - the plasmonic modulator is contacted by a high frequency probe for operation from 225 to 330 GHz (GGB Industries inc.) - the sub-THz signal is coupled to the probe by a Teflon lens, a horn antenna (Flann) and amplified by a medium power amplifier (Frauenhofer) The optical receiver consist of:  - the dual sideband receiver setup discussed in detail in ⁶⁶.
	7. If the authors want to prove that the method proposed in this paper has better cost advantages, sufficient literature comparisons should be provided, including other performance indicators, to highlight the advantages of the paper.
	We thank the reviewer for his questions on more details of the cost advantages and the performance indicators. We believe we answered the second part of this comment already in comment #2 where we provide a detailed comparison to the current state of the art. With respect to the cost-analysis: it is difficult to compare the cost with literature values, as typically this is not a focus on scientific literature. However, we added the following discussion to the supplementary information that outlines the potential cost-advantage of the here proposed technology over conventional solutions relying on UTC PDs.

Supplementary Note 1: Cost Discussions of High-Speed Photodetectors

For sub-THz transmission with signal generation based on opto-electronic converters, there are conventionally uni-travelling carrier photodiodes (UTC-PD) employed. The here proposed high-speed graphene photodetectors could offer a potentially cost-effective alternative with scalable fabrication processes.

The cost of high-speed photonic devices is essentially dominated by:

- (1) Material cost
- (2) Processing cost
- (3) Packaging and co-integration

We outline below along these three points why plasmonic graphene photodetectors could be advantageous over UTC-PD.

(1) Material cost

- UTC PDs are commonly grown on InP wafers. Most epitaxy systems rely on 2-inch or 3-inch wafers, where some systems are also available for 4-inch processing. The cost of the raw InP wafers is in the order of ~ 4 $\$/\text{cm}^2$. The epitaxy growth requires MOCVD/MBE systems that are expensive in procurement as well as in maintenance and in operation. Making a clear estimate on cost per run is difficult but is at least in the order of $\$/\text{cm}^2$.

- Graphene growth on copper foil with a CVD tool results in a cost of 0.019 $\$/\text{cm}^2$ as reported in³⁰. This cost could even further be reduced by re-using the growth substrate after transfer³¹. Transfer can be done on almost arbitrary substrates. Considering a silicon wafer as in this work, the cost for the substrate is ~ 0.4 $\$/\text{cm}^2$.

The cost for the active material stack is thereby expected to be at least an order of magnitude lower than for III-V materials.

(2) Processing cost

Processing the active material to form functional devices is expected to be similar, as both require similar processing technologies. However, handling of Si wafers is typically much easier than handling InP wafers due to lower fragility, higher temperature tolerance, chemical robustness and low toxicity. Additionally, most available equipment is built around Si wafers which makes handling and automatization directly compatible.

(3) Packaging and co-integration

The steps for packaging are expected to have the same cost for both device technologies. However, due to the substrate independence of graphene it is possible to allow for direct co-integration of graphene with e.g. electronic circuits. An example demonstration is the direct integration of graphene with a CMOS camera read-out circuit³². The co-integration of plasmonic devices has also been demonstrated with high-speed electronics³³.

30. Goldsmith, B. R. *et al.* Digital Biosensing by Foundry-Fabricated Graphene Sensors. *Sci Rep* **9**, 434 (2019).

31. Gupta, P. *et al.* A facile process for soak-and-peel delamination of CVD graphene from substrates using water. *Sci Rep* **4**, 3882 (2014).

32. Goossens, S. *et al.* Broadband image sensor array based on graphene-CMOS integration. *Nature Photon* **11**, 366–371 (2017).

33. Koch, U. *et al.* A monolithic bipolar CMOS electronic-plasmonic high-speed transmitter. *Nat Electron* **3**, 338–345 (2020).

8. The amount of data in Figure 4e is a bit small. It is recommended that the authors make more measurements to provide the average value and error range.

We thank the reviewer for this critique and highly appreciated it. We would like to justify the choice on the number of points as follows:

- It has been shown several times that plasmonic-organic modulators as used in this work can provide large bandwidths. Concretely, M. Burla *et al.* (2020) measured flat response up to 500 GHz and a more recent study by Y. Horst *et al.* (2025) found a bandwidth of 1 THz. These measurements show that also with more data points the response remains flat and no significant spikes or drops are observed. The latter work also investigated the impact of various modulator designs on the frequency response and found that all designs result in a simple RC behaviour. Finding a simple RC behaviour only require little data points. Therefore, as the frequency

response of plasmonic-organic modulators are already well documented and backed in multiple experiments, the need and benefit of measuring in finer steps is not apparent to us.

- The in the characzterization setup used optical spectrum analyser for higher frequencies has a resolution bandwidth of 4 GHz. This resolution bandwidth limits the accuracy of the measurements. Therefore, we chose the 5 GHz steps to match our experimental setup.

Yet, motivated by the reviewer's comment, we further investigated how the average value and error range behaves when the number of data points would decrease, i.e., when only a subset of the points is used. We found that the average value and standard deviation value of the measurement set saturate when about 60% of the data is used. See figure below:

Therefore, we extrapolate that additional datapoints would not change the average value or the standard deviation and that the average value and the standard deviation that can be extracted from the data set is precise enough.

We revised the manuscript by motivating the choice of the number of points and by reporting on the error range.

Electro-Optic Conversion with Plasmonic Modulators

[...]

L241: For the modulator in this study, we measured a flat (within a standard deviation of 1.1 dB) frequency response up to 350 GHz as shown in Figure 4e. This is in good agreement with previous frequency response measurements, where flat responses up to 500 GHz and bandwidths of 880 GHz and 997 GHz have been shown^{42,43}. The setup used to measure the frequency response is depicted in Figure 4d.

METHODS:

Frequency modulator characterization

[...]

L414: For small frequencies, a high-resolution OSA (Apex AP2040) was used; for larger frequencies the Yokogawa AQ6370C was used. The frequency response was measured at logarithmically spaced points, with a step size of approximately 5 GHz at the higher end. Finer spacing was avoided, as the OSA's resolution bandwidth is 4 GHz and the frequency response follows a simple RC-response, see Y. Horst et al.⁴³. For calibration, of the electric driving circuit we used sub-harmonic mixers (Virginia Diodes, SAX modules) as discussed above.

We thank the reviewer again for his detailed analysis of the manuscript and his valuable feedback. We hope that by clarifying all open points we could resolve any confusing parts in the manuscript.

Reviewer 2

We thank the reviewer and appreciate their collaborative approach to peer review. We especially value the effort to involve and recognize Early Career Researchers through the Nature Communications co-review initiative. We answered all comments in the respective sections and improved the manuscript according to the constructive feedback provided by the reviewer.

Reviewer 3

The manuscript introduces an all-plasmonic sub-THz wireless link, which enables flexible modulation of the carrier frequency. It also verifies that the previously proposed plasma devices could function as receivers and transmitters, facilitating the conversion of optical and broadband-stabilized terahertz signals. Finally, the authors experimentally demonstrate information transmission over a spatial distance of 5 meters at 285 GHz with data rates of up to 120 Gbit/s. This achievement shows the potential for seamless connection between optical fibers and THz components in short-range applications. While the manuscript provides experimental validation, it lacks sufficient theoretical support.

Furthermore, the devices and methods used in the study are based on previously published work and do not present significant innovations. Therefore, I do not recommend it for publication in a high-level journal such as Nature Communications. The following are some additional comments:

We appreciate the overall feedback provided by the reviewer. We thank the reviewer for pointing out the missing theoretical aspects. We further regret that the novelty aspect of the work has not been communicated clear enough. We address below all of the reviewer's comments in detail to enhance the manuscript.

1. The use of "However" in line 77 appears incorrect. Additionally, the manuscript does not explain why graphene photodetectors (PDs) in the sub-terahertz band have not been previously studied. This omission leaves a gap in the narrative that needs to be addressed.	
	We thank the reviewer for his constructive comment to improve the coherence of the text. Indeed, the use of however here is not unambiguous. We change the text to clarify the sentence. With respect to the comment on why graphene PDs have not yet been studied in the sub-THz-band: Previously it was not possible to use graphene PDs for sub-THz band communication as no graphene PD was yet demonstrated that has (1) sufficient output power while (2) operating at such high frequencies. We highlight this now in the text. The updated section now reads as
	L76: However, Recently, plasmonic photodetectors (PDs) based on graphene have been shown to reach bandwidths beyond 400 GHz ⁴⁶ and even 500 GHz ⁴⁷. With this, the OE side is matching the speed of the EO side and could enable sub-THz wireless communication. However, plasmonic graphene PDs have not been explored or verified for sub-THz communication due to a lack of simultaneously achieving sufficient output powers and operation in the 100s GHz.
2. Figures 1a and 1b lack necessary scale labeling, making it difficult to identify the specific data measurement area discussed in the METHODS section. ...	
	We thank the reviewer for pointing out the missing scale bars. We assume the reviewer is referring to Figure 2a and 2b. We added a scale bar to Figure 2b. In addition, to make it clearer what the dimensions in the METHODS section are referring to we replace the table with a more descriptive text and added a Figure to the supplementary information.
	L348: We used two devices on the same sample for the characterization and data measurements. A first device is used for Fig. 2 and Fig. 3, whereas a second device is used in the experiments in Fig. 5 and Fig. 6. The two devices have the same dimensions. The dimensions of the devices used are as follows: The metamaterial consists of 6 by 10 unit cells, where each unit cell has a size of 1170 nm by 700 nm. Trough the center of the unit cells the interdigitated electrode line is positioned which has a width of 90 nm. In the center, perpendicular to the interdigitate electrode, the plasmonic resonator is positioned which has a length of 246 nm and a width of 90 nm. A visual representation is provided in the Supplementary Information.

Figure S2: Schematic of the plasmonic graphene PD. A visualization of the pad layout, the active area of the device and the metamaterial unit cell dimensions as described in the Methods section is shown. Additionally, a schematic of the layer stack is included.

... Moreover, the manuscript would benefit from providing more detailed parameters and conditions of the laboratory environment, such as humidity and temperature, to ensure reproducibility and clarity.

We thank the reviewer for pointing out the missing information on the measurement conditions. We added the following sentence to the Methods section to clarify the lab conditions:

METHODS:

L376: **Details on experimental environment**

All characterization and wireless data transmission measurements were performed in the same laboratory environment. The laboratory is climate controlled to 22° C +/- 1°C with a relative humidity centered around 50% +/- 5%.

3. Line 125 references supplementary information, but it appears that this material has not been uploaded.

We thank the reviewer for pointing out that the Supplementary Information has been missing. We uploaded the updated file with the revisions.

4. Line 184 mentions an interaction between the RF field and the optical field. Could a more in-depth study and elaboration be given?

We thank the reviewer for this suggestion. We added a study on how the direct conversion from the RF field to the optical field is taking place. We thereby introduce a conversion efficiency and elaborate on the parameter impacting this conversion efficiency.

L188: A broadband Pockels coefficient in the nonlinear material is essential to maintain this broadband performance. Within the modulator's bandwidth, and under null-point biasing, the optical field at the output is given by

$$E = j\alpha(L) \sin\left(\frac{V_{RF}}{V_{\pi L}^{(PS)}} L\pi\right) E_0 e^{j\omega t},$$

where E_0 is the optical input field at frequency ω , $\alpha(L)$ the total optical transmission (dependent on the plasmonic slot length L), $V_{\pi L}^{(PS)}$ the voltage-length-product of one of the phase shifters, and V_{RF} the voltage across the plasmonic slot. The total optical transmission is composed of

$$\alpha(L) = \alpha_{\text{phot}} \alpha_{\text{CPL}} e^{-\beta L},$$

where α_{phot} captures photonic losses (e.g., routing losses, grating coupler losses), α_{CPL} accounts for photonic-plasmonic-photonic coupling loss and $e^{-\beta L}$ is the plasmonic propagation loss. A sinusoidal RF field at frequency Ω_{RF} and with a power of P_{RF} (measured into $R_{50} = 50 \Omega$), yields a peak voltage across the slot of

$$V_{RF} = 2 \cdot \sqrt{2 P_{RF} R_{50}}.$$

Due to the capacitive nature of the load, the signal is reflected, doubling the voltage amplitude compared to the R_{50} load. Applying the Jacobi-Anger expansion to Eq. (1), the fields of the upper (+) and lower (-) sideband read

$$E_{\pm} = j\alpha(L)J_1\left(\frac{V_{\text{RF}}}{V_{\pi\text{L}}^{(\text{PS})}}L\pi\right)e^{j\phi_{\text{RF}}}E_0e^{j(\omega\pm\Omega_{\text{RF}})t},$$

where J_1 denotes the first order Bessel function of the first kind, and ϕ_{RF} the phase of the RF field. Note that the phase information of the RF fields is mapped to the phase of the optical field and the amplitude of the RF field translates to the optical amplitude through the Bessel function. This way, the transmitted complex symbols are mapped to complex symbols in the optical domain. Assuming small-signal conditions, i.e., linearizing J_1 around $V_{\text{RF}} = 0$ V, the intensities of the upper and lower sideband I_{\pm} become

$$I_{\pm} = |\alpha(L)|^2\left(\frac{V_{\text{RF}}}{2V_{\pi\text{L}}^{(\text{PS})}}L\pi\right)^2 \cdot P_{\text{opt}},$$

where $P_{\text{opt}} \propto |E_0|^2$ is the optical input power. Substituting Eq. (3) into Eq. (5), one obtains

$$I_{\pm} = |\alpha(L)|^2\left(\frac{\sqrt{2}R_{50}}{V_{\pi\text{L}}^{(\text{PS})}}L\pi\right)^2 \cdot P_{\text{opt}} \cdot P_{\text{RF}},$$

where the first term is the conversion efficiency η , i.e. $\eta := I_{\pm}P_{\text{opt}}^{-1}P_{\text{RF}}^{-1}$ with $[\eta] = \text{W}^{-1}$. This conversion efficiency measures the sideband power for a given RF power P_{RF} and optical power P_{opt} . The conversion efficiency depends on the cross section of the plasmonic waveguide and the nonlinear material through $V_{\pi\text{L}}^{(\text{PS})}$ and β , and on the slot length through $|\alpha(L)|^2L^2$. Taking the cross-section geometry, i.e., $V_{\pi\text{L}}^{(\text{PS})}$ and β , as fixed, the slot length L^* that maximizes the conversion efficiency is the characteristic decay length of a (surface) plasmon polariton, i.e.,

$$L^* = \frac{1}{\beta}.$$

Further details can also be found in Reference ⁵⁵. The conversion efficiency then reads

$$\eta = \frac{\pi^2}{e^2}R_{50} \cdot \left(\frac{|\alpha_{\text{Phot}}\alpha_{\text{CPL}}|}{V_{\pi\text{L}}^{(\text{PS})}\beta}\right)^2.$$

From Eq. (8), it becomes clear that the conversion efficiency is large when the losses of the coupling and photonic parts are small and the $V_{\pi\text{L}}\beta$ of the plasmonic cross section is small. The $V_{\pi\text{L}}$ can be approximated by

$$V_{\pi\text{L}}^{(\text{PS})} = \lambda \cdot \frac{1}{n_{\text{mat}}^3 r_{\text{eff}}} \cdot \frac{w_{\text{slot}}}{\Gamma},$$

where λ is the wavelength, n_{mat} and r_{eff} the refractive index and the effective Pockels coefficient of the nonlinear material, respectively, w_{slot} is the slot width and Γ is the interaction coefficient ⁵⁶. The material properties for organic materials can reach up to $n_{\text{mat}}^3 r_{\text{eff}} = 8700 \text{ V}^{-1}\text{pm}$ ⁵⁷. The interaction coefficient Γ takes into account how much the effective refractive index changes compared to the bulk material change upon applied voltages. For plasmonic waveguides, the interaction coefficient Γ can be close to 1, as there is a good overlap between the RF and optical fields ^{54,56,58}. Furthermore, plasmonic waveguides can have uniquely narrow slot width yielding ultra-low, in-device $V_{\pi\text{L}}^{(\text{PS})}$ down to $150 \text{ V}\mu\text{m}$ ⁵⁹. While narrowing the slot reduces $V_{\pi\text{L}}^{(\text{PS})}$, it increases plasmonic losses β . Interestingly, simulations show that the product $V_{\pi\text{L}}^{(\text{PS})}\beta$ continues to decrease with narrower slots ⁵⁸. However, excessively narrow slots degrade organic nonlinearity, making $w_{\text{slot}} \approx 100 \text{ nm}$ a practical optimum ⁶⁰. Assuming negligible photonic losses ($|\alpha_{\text{Phot}}|^2 = 1$) and coupling loss $|\alpha_{\text{CPL}}|^2 \triangleq 2 \cdot 0.5 \text{ dB}$ and $V_{\pi\text{L}}^{(\text{PS})}\beta \triangleq 100 \text{ VdB}$ ⁵⁸, the achievable conversion efficiency is around -35 dBm.

L249: The peak-to-peak voltage required for switching the MZM from the minimum to the maximum transmission (V_{π}) is 11.1 V. However, due to the lumped-capacitor nature of the plasmonic modulator,

	see Eq. (3), the voltage induced by an incoming RF signal at a certain power is twice that of a conventional 50 Ω terminated modulator. This voltage enhancement leads to an effective V_{π} of 5.5 V, highlighting the advantage of plasmonic modulators over their 50 Ω counterparts. With those values, the conversion efficiency, see Eq. (8), is -42.1 dBm.
5.	Most importantly, as stated in the paper, the plasma devices used have all been published, so are there any different effects or innovations in the devices used in this paper? Moreover, it is suggested that some corresponding physical mechanisms or details be added to this paper as well to improve the readability and novelty of the article.
	We are grateful for the reviewer's feedback on the missing discussions. We assume the reviewer is referring to the plasmonic devices here. With respect to the physical working principle of the plasmonic modulator, we made the changes discussed above on the reviewer's comment (4.) With respect to the physical working principle of the plasmonic graphene photodetector, we made the following changes to the main text.
I109	(added a reference): A metamaterial perfect absorber (N. Landy, 2008) (MPA) layer stack in the form of a horizontal metal-insulator-metal is used.
I114:	We explain the physical operation principle that results from the above-described architecture. The plasmonic graphene PD operates in a photovoltaic (PV) mode, which is enabled by the metamaterial perfect absorber structure. Light impinging on the metamaterial structure leads to a plasmonic resonance at the resonator. The resonator is inductively coupled to the metallic backplane, which leads to a trapping of the light within the layer stack ^{49,50}. A fraction of the light is absorbed within the graphene layer close to the resonators. The absorbed photons generate free charge carriers. Due to contact doping at the resonators ⁵¹, a built-in field is induced, which drives the charge carriers to the collection electrodes and generates the PV photocurrent.
I126:	When applying a gate voltage V_G with the gate needle, we can modify the doping within the graphene channel and thereby its resistance and the shape of the built-in field.
	Furthermore, we added the following changes to highlight the changes to the previous published work on the PD.
I164:	The maximum extracted photocurrent is 0.556 mA, which, to the best of our knowledge, is the highest photocurrent reported for any high-speed zero-bias graphene photodetector as summarized in Figure 3c. The changes which lead to the improvements in responsivity and maximum photocurrent compared to the prior work are further discussed in the Method section. With the here reported values the device can be used in any scenario requiring high-speed infrared photodetectors, for example data communication. Next, we will verify the usability of the photodetector as a sub-THz source in a wireless data transmission scenario with a carrier frequency of 285 GHz in the following sections after discussing the sub-THz receiver in the form of a plasmonic modulator.
	METHODS: I357: Plasmonic photodetector design As outlined in the main text, the metamaterial design follows a similar architecture as in [46]. To achieve the factor x4 improvement in responsivity and the factor x12 higher output photocurrent, the following design considerations were taken into consideration. The plasmonic graphene PD in operation is modelled as an equivalent circuit consisting of  • The resistance of the interdigitated electrodes including the resonators • The resistance of the graphene channel between two interdigitated electrodes • The resistance of the contact between the interdigitated electrodes and the graphene

- The current source representing the photocurrent
- The load resistance to which the PD delivers the current.

These values per unit cell are calculated from the physical parameters of the metamaterial structure, i.e., unit cell size, interdigitated electrode width and height. The parameters were optimized in a multi-dimensional optimization scheme where the optical absorption given by the metamaterial, the RF power delivered to the load resistance and the optical power distribution were considered. We note that the electrical design of the metamaterial cannot be arbitrarily changed without influences the optical resonant behavior of the metamaterial. Therefore, such a multi-dimensional optimization approach is required.

6. In line 216, it should be figure5, not figure4figure5.

We appreciate the reviewer for catching this typo. We fixed it in the updated version of the manuscript.

7. The authors claim superior performance for their work. To substantiate these claims, it is suggested that a detailed table comparing key metric parameters with existing literature be included. This would provide a clearer benchmark for evaluating the manuscript's contributions.

We thank the reviewer for his interest in an up to date comparison to state of the art. We added the corresponding table to the supplementary information. Furthermore, we added a Figure 3c that graphically represents the key properties (high bandwidth and high photocurrent) required for the sub-THz communication link. The changes look as follows:

Figure 3: DC photocurrent and RF power generation of plasmonic graphene photodetectors.
a) Generated DC photocurrent as function of optical power on the device. The slope leads to a responsivity of 6 mA/W. Gray dots correspond to measurements performed with a lock-in amplifier. Blue to green squares correspond to measurements where the PD was connected to the electrical spectrum analyzer and sub-harmonic mixers. b) Generated RF power as function of DC photocurrent for three RF frequencies. 70 GHz: blue, 140 GHz blue-green, 280 GHz green. c) Comparison of reported high-speed graphene photodetectors as function of achieved bandwidth and maximum output photocurrent. Detailed parameters for each data point can be found in supplementary table 1.

Ref.	Year	Bandwidth	Max. Output	Device Resistance	Det. Mech.	Responsivity	Bias Voltage	Gate Voltage	
1	2009	>40 GHz	---	a	~2000 Ω	PV	0.5 mA/W	0 V	80 V
2	2010	16 GHz	~63 uA	b	140 Ω	PV	1.5 mA/W	0 V	-15 V
						-	6.1 mA/W	0.4 V	-15 V
3	2013	>20 GHz	~19.2 uA	c	---	PV	15.7 mA/W	0 V	No Gate
4	2013	18 GHz	~2.6 uA	-	---	PV	50 mA/W	0 V	No Gate
5	2014	3 GHz	~240 uA	-	~100 Ω	PC	57 mA/W	0.4 V	9 V
6	2014	41 GHz	~20 uA	-	187 Ω	PV	16 mA/W	0 V	No Gate
7	2015	42 GHz	~0.2 uA	b	98 Ω	PTE	78 mA/W	0 V	2.9 V
8	2016	65 GHz	~1 mV ~15 uA	a	254 Ω	PTE	3.5 V/W 35 mA/W	0 V	~6 V, ~6 V
						---	76 mA/W	0.3 V	~6 V, ~6V
9	2017	76 GHz	15.9 uA	a	130 Ω	BOL	1 mA/W	1 V	No Gate
10	2018	>18 GHz	1.5 mV 7.5 uA	x	200 Ω	PTE	4.7 V/W 48 mA/W	0 V	4 V, -2 V
						PC	170 mA/W	0.4 V	-8 V, 5 V
11	2018	> 50 GHz	3.7 uA	c	68 Ω	PC	1050 mA/W	0.02 V	-20 V
12	2018	>110 GHz	274 uA	b	100 Ω	BOL	400 mA/W	-0.6 V	No Gate
						PTE/PV	3 mA/W	0 V	No Gate
13	2018	38 GHz	~5.5 uA	-	300 Ω	---	0.57 mA/W	0 V	No Gate
14	2019	42 GHz	2.9 mV ~1.83 uA	x	~2000 Ω	PTE	12.2 V/W	0 V	-4 V, -6 V
15	2019	>110 GHz	144 uA	d	100 Ω	PV/PC	360 mA/W	2.2 V	No Gate
16	2019	>70 GHz	15 uA	a	---	PC	300 mA/W	0.5 V	No Gate
17	2020	>67 GHz	4 mV 8 uA	a	~500 Ω	PTE	6 V/W 12 mA/W	0 V	3 V, -1 V
18	2020	>40 GHz	~87 uA	b	~150 Ω	BOL	396 mA/W	0.3 V to 1 V	0 to 2.8 V
19	2021	70 GHz	15.1 mV ~214 uA	b, x	~70 Ω	PTE	3.5 V/W	0 V	-1 V, -8 V
20	2021	70 GHz	208 uA	a	---	PC	104 mA/W	0.8 V	No Gate
21	2021	>67 GHz	---	a	172 Ω	PC	100 mA/W	0.3 V	No Gate
22	2021	12 GHz	38.6 mV 32.2 uA	b, x	~1200 Ω	PTE	90 V/W 75 mA/W	0 V	-0.5 V, -2.1 V
23	2022	78 GHz	233 uA	b	231 Ω	BOL	603.9 mA/W	1.9 V	No Gate
24	2023	30 GHz	0.21 uA	-	355 Ω	IPE/PTFE	0.33 mA/W	6 V	No Gate
25	2023	>500 GHz	45.8 uA	-	235 Ω	PV	1.57 mA/W	0 V	-2.5 V
26	2024	36 GHz	142 uA	b	150 Ω	PC	460 mA/W	0.5 V	No Gate
27	2024	420 GHz	18.5 mV 5.4 uA	b	5806 Ω	PTE	1.6 V/W 0.4 mA/W	0 V	0.5 V
28	2025	>14 GHz	17 uA	b	286 Ω	BOL	122 mA/W	2.6 V	No Gate
29	2025	155 GHz	---	a	255 Ω	PV/PTE/PC	68 mA/W	0.3 V	-38 V
This work	2025	>330 GHz	556 uA	-	22 Ω	PV	6 mA/W	0 V	-6.75 V

8. *Several details in the text need to be reviewed for greater rigor, as previously highlighted in points 1, 3, and 6.*

We thank the reviewer for his comment. We went through the whole manuscript and made sure that, in addition to the above answer, formulations have been clarified and additional references have been added where needed.

We appreciate the detailed comments of the reviewer and his effort to help us improve our manuscript. The manuscript is now adapted accordingly by addressing all comments in detail and adding the lacking discussions on physical working principles, details on the experiments and an updated state-of-the-art comparison.